**JCB** Journal of Cell Biology

**TOOLS**

# Murine glial protrusion transcripts predict localized *Drosophila* glial mRNAs involved in plasticity

Jeffrey Y. Lee[1,2]*, Dalia S. Gala[2]*, Maria Kiourlappou[2], Julia Olivares-Abril[2], Jana Joha[2], Joshua S. Titlow[2], Rita O. Teodoro[3], and Ilan Davis[1,2]

The polarization of cells often involves the transport of specific mRNAs and their localized translation in distal projections. Neurons and glia are both known to contain long cytoplasmic processes, while localized transcripts have only been studied extensively in neurons, not glia, especially in intact nervous systems. Here, we predict 1,740 localized *Drosophila* glial transcripts by extrapolating from our meta-analysis of seven existing studies characterizing the localized transcriptomes and translatomes of synaptically associated mammalian glia. We demonstrate that the localization of mRNAs in mammalian glial projections strongly predicts the localization of their high-confidence *Drosophila* homologs in larval motor neuron–associated glial projections and are highly statistically enriched for genes associated with neurological diseases. We further show that some of these localized glial transcripts are specifically required in glia for structural plasticity at the nearby neuromuscular junction synapses. We conclude that peripheral glial mRNA localization is a common and conserved phenomenon and propose that it is likely to be functionally important in disease.

## Introduction

It is well established that specific mRNAs are actively transported and localized to distal compartments of neurons, including those encoding cytoskeletal proteins, neurotransmitters, membrane proteins, and ribosomes. In these studies, mRNA transport and local translation have been proposed as mechanisms for local regulation of synaptic plasticity, which is the process underlying learning and memory (Biever et al., 2019; Daniel and Shumer, 2017; tom Dieck et al., 2015; Holt et al., 2019; Rangaraju et al., 2017; Smith et al., 2001; Sutton and Schuman, 2006; Terenzio et al., 2017; Wang et al., 2009, 2010)

Asymmetric mRNA localization is likely to be as important in glia as it is in neurons but has received less attention in glia. Like neurons, glial cells display diverse morphologies ranging from immune-cell-like microglia to elongated myelinating oligodendrocytes and astrocytes that contact thousands of synapses. Therefore, astrocytes are thought to integrate information from distinct neuronal populations (Allen and Lyons, 2018; Jessen et al., 2015; Ransom and Ransom, 2012; Volterra and Meldolesi, 2005). Astrocytes also perform key roles in the uptake and release of neurotransmitters and the maintenance of ionic balance. Glia form the blood–brain barrier that regulates neuronal metabolism and local solute homeostasis, as well as the secretion of gliotransmitters and mediation of signaling pathways (De Pittà et al., 2016; Ota et al., 2013; Wang et al., 2022).

The function of mRNA localization in astrocytes is being actively investigated (Boulay et al., 2017; Mazaré et al., 2020; Oudart et al., 2020; Sapkota et al., 2022). Oligodendrocytes are thought to modulate synaptic plasticity via myelination (Bacmeister et al., 2020; de Faria et al., 2021; Fields, 2005, 2008; Fields and Bukalo, 2020; Pan et al., 2020) and are one of the glial cell types where the importance of mRNA localization has been extensively studied, leading to several groundbreaking discoveries of numerous transport mechanisms and localization elements (Ainger et al., 1997; Carson et al., 2008; White et al., 2008; Yergert et al., 2021). Microglia can preferentially phagocytose synaptic endings and modulate connectivity in the visual cortex (Andoh and Koyama, 2021; Graeber, 2010; Morris et al., 2013; Schafer et al., 2012; Vasek et al., 2023). All of these functions occur in remote cellular processes that are separated from the core gene expression machinery in the cell body. These varying glial morphologies and functional roles are suggestive of mRNA localization to glial protrusions, and indeed some research has outlined its importance (Blanco-Urrejola et al., 2021; Meservey et al., 2021). However, there has yet to be a systematic study comparing glial protrusion datasets to assess whether mRNA localization is pervasive in glia or the extent to which localized transcripts overlap in the periphery of different glial subtypes or in

[1]School of Molecular Biosciences, College of Medical, Veterinary and Life Sciences, University of Glasgow, Glasgow, UK;   [2]Department of Biochemistry, University of Oxford, Oxford, UK;   [3]iNOVA4Health, NOVA Medical School | Faculdade de Ciências Médicas, Universidade Nova de Lisboa, Lisboa, Portugal.

*J.Y. Lee and D.S. Gala contributed equally to this paper.   Correspondence to Ilan Davis: ilan.davis@glasgow.ac.uk.



neurons. It is also not known whether such glial mRNA localization is conserved and functionally important.

To address these questions, we have carried out a meta-analysis of, to our knowledge, all the published mammalian glial peripheral or synaptic transcriptomes and translatomes. We have emphasized the presence of transcripts in the distal periphery rather than enrichment when defining whether mRNAs are localized to the periphery, given the large distance of the peripheral cytoplasm from the cell body. This way of defining mRNA localization allows us to consider the different libraries uniformly and combine all the data sets together. We found several classes of mRNAs that are localized in multiple glial subtypes and determined a common set of 5,028 transcripts present in the majority of the analyzed libraries, representing multiple types of mammalian glial cells. We used this cohort to find high-confidence *Drosophila* homologs and filtered the list for glial expression using the Fly Cell Atlas data, a single-cell nuclear sequencing atlas derived from entire adult flies (Li et al., 2022). Using the single-cell dataset, we selected transcripts expressed in glial subtypes that create distal protrusions in the larval NMJ or make close connections to the muscle boundary. Specifically, we chose perineural and subperineural glia as well as wrapping glia, which are continuous with the ensheathing glia in the peripheral nervous system (PNS) (Sepp and Auld, 2003; Yildirim et al., 2019), resulting in 1,740 mRNAs predicted to be localized in these three glial subtypes.

To test our predictions of mRNA localization in *Drosophila* glial projections, we studied in greater detail 15 conserved transcripts, which we discovered, in an unrelated survey of 200 transcripts across the nervous system, were likely to be localized at glial distal projections near the NMJ axonal synapses (Titlow et al., 2023). We first confirmed that these 15 transcripts were indeed localized using definitive glial markers and more extensive, higher precision 3D microscopy. We then assessed whether these experimentally determined localized transcripts were also predicted as localized in our meta-analysis. We found that >70% (11 of the 15) conserved localized transcripts were correctly predicted as localized at the protrusions, through being in the list of 1,740 transcripts. Furthermore, in follow-up experiments, we tested whether the localized transcripts are required in the glia for the correct plasticity of the adjacent neuronal synapses using a spaced potassium stimulation assay (Ataman et al., 2008; Roche et al., 2002). We found that 5 of the 15 localized transcripts impaired synaptic plasticity in the neighboring wild type neurons, reducing their ability to make new synaptic boutons. Our results suggest that mRNA localization to protrusions represents an important and conserved mechanism that allows glia to rapidly regulate the plasticity of adjacent distal axonal synapses.

## Results

### Collation of multiple studies characterizing transcript localization to peripheral glial cytoplasmic projections

To identify a common set of localized glial transcripts, we performed a meta-analysis of, to our knowledge, all published mammalian localized and synaptic transcriptomic datasets

derived from multiple types of glia (Azevedo et al., 2018; Boulay et al., 2017; Mazaré et al., 2020; Sakers et al., 2017; Thakurela et al., 2016; Thomsen et al., 2013; Vasek et al., 2023) displayed in Table 1. These glial cell types included four studies of astrocytes and one study each of oligodendrocytes, microdissected myelin, and microglia. The seven studies used various methods of purifying localized transcripts, which are summarized in Table 4 and Fig. 1 A. We collated seven transcriptomic and four translating ribosome affinity purification (TRAP) libraries using a standardized pipeline for interoperability between the 11 datasets (Fig. 1 B). For details of how our analysis was performed and how we created a single data framework from all these independently acquired data sets, see the Data meta-analysis methods section of the Materials and methods.

Using our bespoke data processing pipeline, we found 5,028 common transcripts detected in at least seven libraries of localized mammalian glial transcripts. We tested whether this population of localized mRNAs is unique to the periphery of glia or whether they are also shared with the neurite transcriptome. To distinguish these possibilities, we compared the combined list of glial localized transcripts to the group of transcripts shown to commonly localize to the mammalian neuronal projections (von Kügelgen and Chekulaeva, 2020). We found a very significant overlap between the transcripts present at the periphery of both glia and neurons (Fig. 1 D). To further investigate the generality of localized transcripts, we intersected our dataset with the protrusion transcriptome of non-neural breast cancer cell line (MDA-MB231) that makes highly protrusive cell projections (Mardakheh et al., 2015). We found that more than a thousand genes are commonly localized in all glia, neuron, and cancer cell protrusions (Fig. 1 E). Our observations corroborate recent studies highlighting the conserved mRNA transport mechanisms between cell types (Goering et al., 2022, *Preprint*) and significant overlap of localized transcripts within astrocytes, radial glia, and neurons (D'Arcy and Silver, 2020).

### Meta-analysis of localized glial transcriptomes yields a predicted group of transcripts localized to the periphery of *Drosophila* peripheral glia

To address whether the mammalian glial localized transcripts were likely to be conserved across higher eukaryotes, in a specific case that is highly experimentally tractable, we intersected our mammalian dataset with the glial subtypes that extend their processes to the PNS of *Drosophila*, a well-established model for studying mRNA transport and local translation. We first filtered the 5,028 mammalian genes to only include the ones that have high-confidence homologs, retaining genes with a DIOPT score of at least 8 as a cutoff (Fig. 1 C). This filtering process resulted in a list of 3,606 *Drosophila* genes that we used to query the glial nuclear transcriptome from the Fly Cell Atlas (Li et al., 2022) (Fig. 1 F). We chose transcripts that were expressed in perineurial glia (PG) and subperineurial glia (SPG), the glial subtypes known to extend their processes to the *Drosophila* neuromuscular junction (NMJ) (Fig. S1, A and B). These glia have very long cytoplasmic extensions and their cell bodies can be hundreds of micrometers away from their furthermost projections (Brink et al., 2012; Sepp et al., 2000, 2001; Sepp and Auld,

Table 1. **Summary of all the available localized and synaptic glial transcriptomic datasets which were analyzed in this study**

| Reference | Cell type | Specimen type | Type of data | Organism |
|---|---|---|---|---|
| Azevedo et al., 2018 | Oligodendrocytes protrusions (OPCs) | Primary cultures of OPCs from rat brain in Boyden Chamber | Soma and protrusion transcriptome | Rat |
| Thomsen et al., 2013 | Perisynaptic astrocytic processes (PAPs) | Type 2 astrocyte mouse cell line C8-S in Boyden Chamber | Soma and protrusion transcriptome | Mouse |
| Thakurela et al., 2016 | Mouse CNS myelin (oligodendrocytes) | Biochemically purified myelin from whole brains of male c57Bl6/N mice | Myelin transcriptome at various developmental stages (P18, P75, 6 and 24 months) | Mouse |
| Boulay et al., 2017 | Astrocyte endfeet | Mechanically isolated brain vessels with attached astrocytic endfeet | Soma and protrusion translatome | Mouse |
| Sakers et al., 2017 | Perisynaptic astrocytic processes (PAPs) | Isolated astrocyte-GFP-tagged ribosome-bound RNA from the synaptoneurosome fraction from homogenized cortices (including hippocampi) | Soma and protrusion transcriptome and translatome | Mouse |
| Mazaré et al., 2020 | Perisynaptic astrocytic processes (PAPs) | Isolated astrocyte-GFP-tagged ribosome-bound RNA from synaptogliosomes from dorsal hippocampi | Soma and protrusion translatome | Mouse |
| Vasek et al., 2023 | Peripheral Microglia Processess (PEMPs) | GFP tagged ribosomes from Peripheral Microglia Processess | Protrusion translatome | Mouse |

2003) (Fig. 1 G). We also included ensheathing glia (EG) as they are the closest glial cell type in the Fly Cell Atlas that corresponds to the wrapping glia (WG) of the PNS. The ensheathing glia in the CNS are continuous with the wrapping glia in the PNS and express similar marker genes such as *nrv2* (Yildirim et al., 2019) (Fig. S1 C). Our filtering of the mammalian glial localized mRNAs with high confidence *Drosophila* homologs by expression in any of the three *Drosophila* PNS glial subtypes (PG, SPG, and WG/EG) yielded a group of 1,740 transcripts that we classify as "predicted to be present" in the projections of the *Drosophila* PNS glia. We hereafter refer to these transcripts as glia protrusion-localized transcripts.

To explore the relevance of the cell type to the 1,740 localized transcripts identified, we intersected this group with the bulk transcriptomics study of perisynaptic Schwann cells (PSCs), FACS-purified from mice (Castro et al., 2020). PSCs are non-myelinating glia associated with the NMJ that can influence synaptic activity and plasticity of adjacent motor neurons, and therefore this cell type is closely related to the three glial types that make close contact with fly motoneurons. We found that 97% of the 1,740 genes are expressed in the PSC, which suggests our group of localized transcripts may be highly relevant for the cell-type specific function of synaptic glia (Fig. 2 A). To further investigate whether RNA localization is a selective process in relation to function, we assessed the localization pattern of housekeeping genes in glial protrusion versus the whole cell (Joshi et al., 2022). GO analysis revealed that transcripts encoding proteins with housekeeping functions were significantly under-represented in the glial-localized 1,740 group compared with the background (Fig. 2 B). Taken together, these results indicate a degree of functional selectivity of glial protrusion localized transcripts.

**Gene ontology analysis reveals the statistical enrichment of transcripts related to mRNA trafficking, membrane composition, and cytoskeletal regulation**
To understand the functional characteristics of this group of localized transcripts in more detail, we identified enriched Gene

Ontology (GO) terms within the localized 1,740 genes (Fig. 3 A). A table with the full GO analysis is provided in Table S2. Our analysis revealed that the localized transcripts are enriched for biological processes involving morphogenesis/development, subcellular localization, cytoskeletal organization, signaling processes, and mRNA metabolism (Fig. 3 A). In terms of molecular function, RNA-binding, cytoskeleton-binding, as well as transmembrane transporter genes were highly enriched in our gene set. Finally, our analysis of GO terms for the cellular component showed enrichment of terms related to vesicle/membrane transport, ribonucleoprotein granules, and ribosomes in connection with the cytoskeleton, and the synapse, including synaptic terminal junctions.

To assess whether the localized 1,740 genes represent genes with specific sets of functions, we performed a comparative GO enrichment analysis between our gene set versus genes that are expressed in the three glial subtypes but are not predicted to localize to glial protrusions. Our analysis revealed that non-localized genes are enriched for GO terms unrelated to localized transcripts or reduced enrichments of overlapping GO terms (Fig. S2 A). This result indicates that the 1,740 transcripts likely perform functions related to their localization compartment distinct from non-localized RNAs. Then, we also asked whether locally translated transcripts could be functionally distinct from localized RNAs. To address this question, we further filtered localized transcripts for evidence of translation in glial protrusions in all four TRAP-seq libraries in mammals, resulting in 778 genes (three astrocytes and one microglia, Fig. 1 B). We found a comparable distribution of enriched GO terms for localized versus localized and translated genes (Fig. S2 B), suggesting the subset of locally translated transcripts are functionally similar to the localized RNAs. Taken together, our GO enrichment analysis highlights a significant over-representation of functions related to membrane trafficking, cytoskeleton regulation, local translation, and cell–cell communication within the localized transcripts, all of which are likely to be very active at the distal periphery of polarized cells.

Figure 1. **Conserved transcripts localized to glial protrusions. (A)** Graphical representation of the techniques used to separate the protrusion-localized transcripts in each of the studies analyzed here. **(B)** Bar graph showing the datasets included in this study. The high confidence expression cut-off was set at

TPM >10, and transcripts detected in more than three independent datasets are shown in orange color. **(C)** Identification of high-confidence *D. melanogaster* orthologs of 5,028 mammalian genes that were detected in at least 7 datasets. DRSC Integrative Ortholog Prediction Tool (DIOPT) score of 8 was used as a cut-off. **(D)** Comparison of transcripts localized in the glial protrusion and neurites (von Kügelgen and Chekulaeva, 2020). **(E)** Upset plot showing a comparison of cellular protrusion localized transcripts between glia, neuron, and breast cancer cell line (MDA-MB231) (Mardakheh et al., 2015). **(F)** t-SNE plot of combined glial single-nuclei RNA-seq from the Fly Cell Atlas (Li et al., 2022). Perineurial, subperineurial, and ensheathing glial cell clusters are depicted in orange-red colors. **(G)** A schematic representing the distribution and location of glial cells in the *Drosophila* third instar larva. Six subtypes of glia exist in the larvae: perineurial and subperineurial glia, cortex glia, astrocyte-like and ensheathing glial cells and, finally, the peripheral nervous system (PNS) specific wrapping glia (Yildirim et al., 2019). Ensheathing glia of the CNS are continuous with the wrapping glia of the CNS. The perineurial and subperineurial glia have long and extensive projections which reach all the way to the neuromuscular junction (NMJ).

To determine whether the 1,740 localized glial transcripts are enriched in particular signaling and enzymatic pathways, we carried out a reactome-pathway enrichment analysis. Here, we found a significant enrichment of terms related to signaling pathways, such as Hedgehog or Wnt pathways, as well as terms related to mRNA translation, mRNA stability regulation and nonsense-mediated decay (Fig. 3 B). Similar to the GO analysis described above, we found a considerably reduced enrichment of these reactome terms in genes that were not predicted to localize to the glial periphery (Fig. S2 C). These significant associations suggest that many of the localized mRNAs at the periphery of glia are required for neuron–glia, glia–muscle, or glia–glia communication, and localized mRNA processing and metabolism at the distal cytoplasm of glia. Such mechanisms could potentially include non-canonical mRNA processing that has been previously suggested to be involved in brain physiology and cancer (Pitolli et al., 2022).

### The 1,740 transcripts predicted to localize at the glial periphery are enriched for neurodegenerative and neuropsychiatric disorder-associated genes

Neurodegenerative and neuropsychiatric disorders have been associated with the disruption of a number of post-transcriptional mechanisms, such as local translation, RNA-binding protein

(RBP) activities, and the formation of RNA-rich granules (Blanco-Urrejola et al., 2021). To determine whether our predicted peripherally localized *Drosophila* glial transcripts are statistically enriched in previously studied nervous system disorders, we carried out a disease ontology analysis. We found a significant enrichment of disease terms related to neuro-pathologies in our set of 1,740 glial protrusion-localized transcripts (Fig. 4 A). Notably, terms related to neurodegeneration and dementia were particularly enriched in the 1,740 genes and were also found in previous studies to be connected to glial-related mechanisms that cause the diseases (Blanco-Urrejola et al., 2021). Importantly, we found the majority of the disease relations were lost or significantly under-enriched when we performed the same analysis with genes that were not predicted to localize to protrusions (Fig. 3 B). This result highlights the unique level of association of neurological disease with genes whose transcripts are localized in glial protrusions compared with non-localizing RNAs. We also compared localizing and non-localizing transcripts for their intersection with the SFARI gene database, which is a well-annotated list of genes that have associations with autism spectrum disorder (https://www.sfari.org/). Although we found that both localized and non-localized transcripts showed a higher than chance level of overlap with the ASD-associated list of genes, we found orders of magnitude

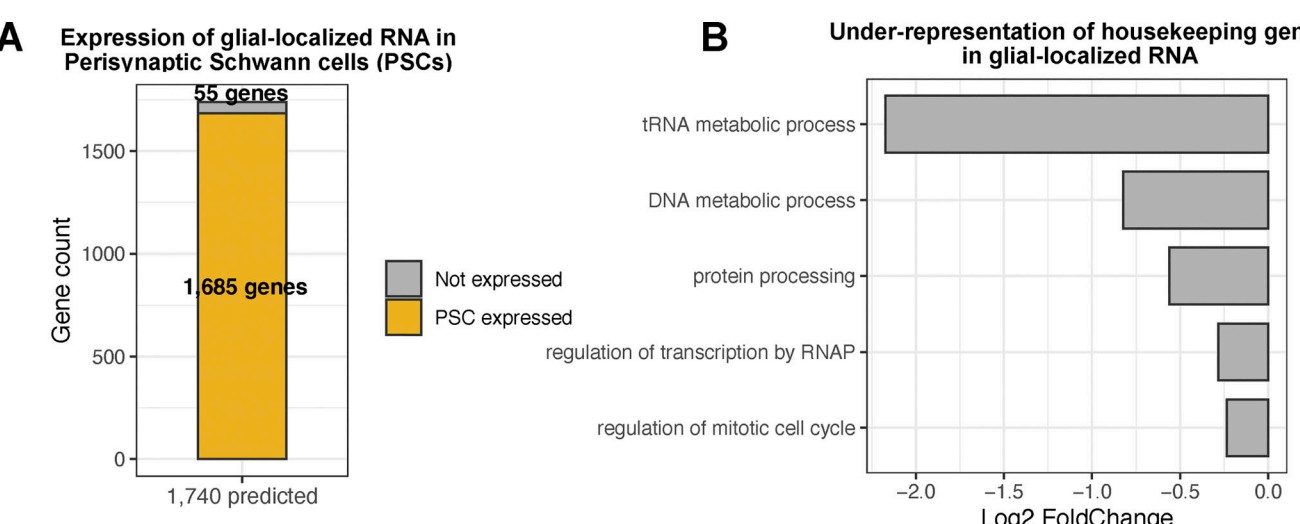

Figure 2. **Functional selectivity of glial protrusion-localized transcripts. (A)** Comparison of glial protrusion-localized transcripts (1,740 genes) with the mouse Perisynaptic Schwann Cell (PSC) transcriptome (Castro et al., 2020). For PSC RNA-seq data, TPM >10 cutoff was used for RNA expression and the recovered genes were converted to fly genes with DIOPT score ≥8 cutoff. **(B)** Under-representation of housekeeping genes in the glial protrusion-localized transcripts. Housekeeping Gene Ontology (GO) terms were selected from (Joshi et al., 2022), and the enrichment or under-representation of each term was assessed using a hypergeometric test.

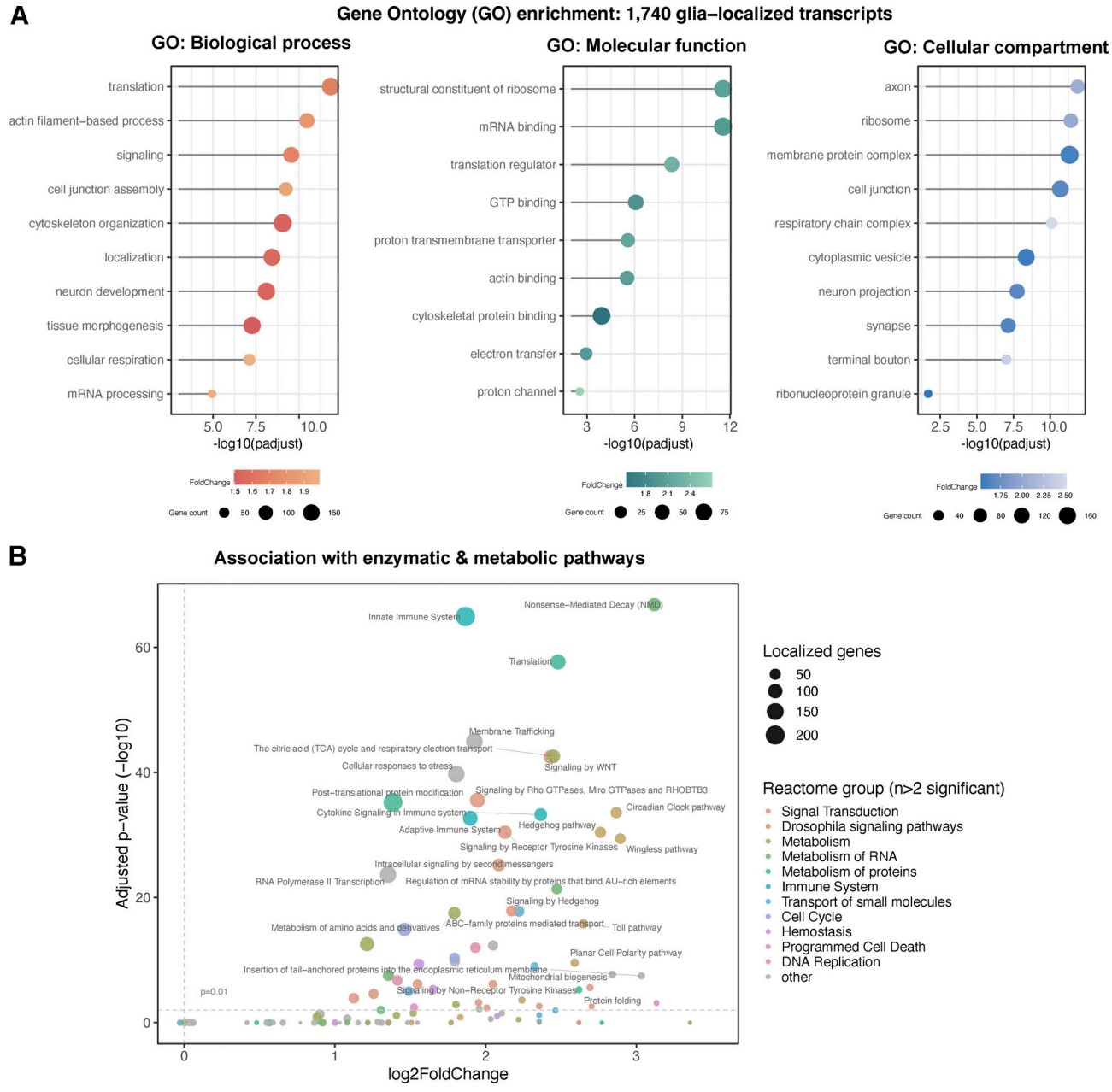

Figure 3. **Functional annotation of glial protrusion-localized transcripts. (A)** Overviews of gene ontology (GO) terms enriched in the 1,740 glial protrusion-localized transcripts in *Biological Process*, *Molecular Function* and *Cellular Component* categories. Enriched GO terms (FoldChange >1.5, adjusted P value <0.01) were simplified using the semantic similarity algorithm. *D. melanogaster* genes with high-confidence orthologues to *M. musculus* genes (DIOPT score ≥8) were used as background. Full GO enrichment analysis table is given in Table S2. **(B)** Reactome pathway enrichment analysis of the 1,740 glial protrusion-localized transcripts against the whole *D. melanogaster* transcriptome as background. The size of the points shows the number of localized genes corresponding to the term and the color of the point indicates the parent term in the Reactome pathways hierarchy. Enrichment was assessed with a hypergeometric test and corrected for multiple hypothesis testing.

higher significance of the overlap with the localized transcripts versus non-localized (Fig. 4 C). Interestingly, we found a more significant degree of overlap between the SFARI gene list and glial localized transcripts compared with the neurite-localized transcripts (von Kügelgen and Chekulaeva, 2020) (Fig. 4 D). Although the absolute number of overlapping genes was higher between neurite and SFARI list, the lower significance may be due to the much higher number of transcripts localized in

neurites versus glial protrusions. We conclude that localized genes in both cell types are highly relevant for the pathology of ASD. Overall, these results support the idea that mRNA localization in glia and the specific list of 1,740 genes we have highlighted are enriched in genes associated with neurological disorders and are more likely to be involved in the etiology of these diseases than those that do not localize to glia periphery.

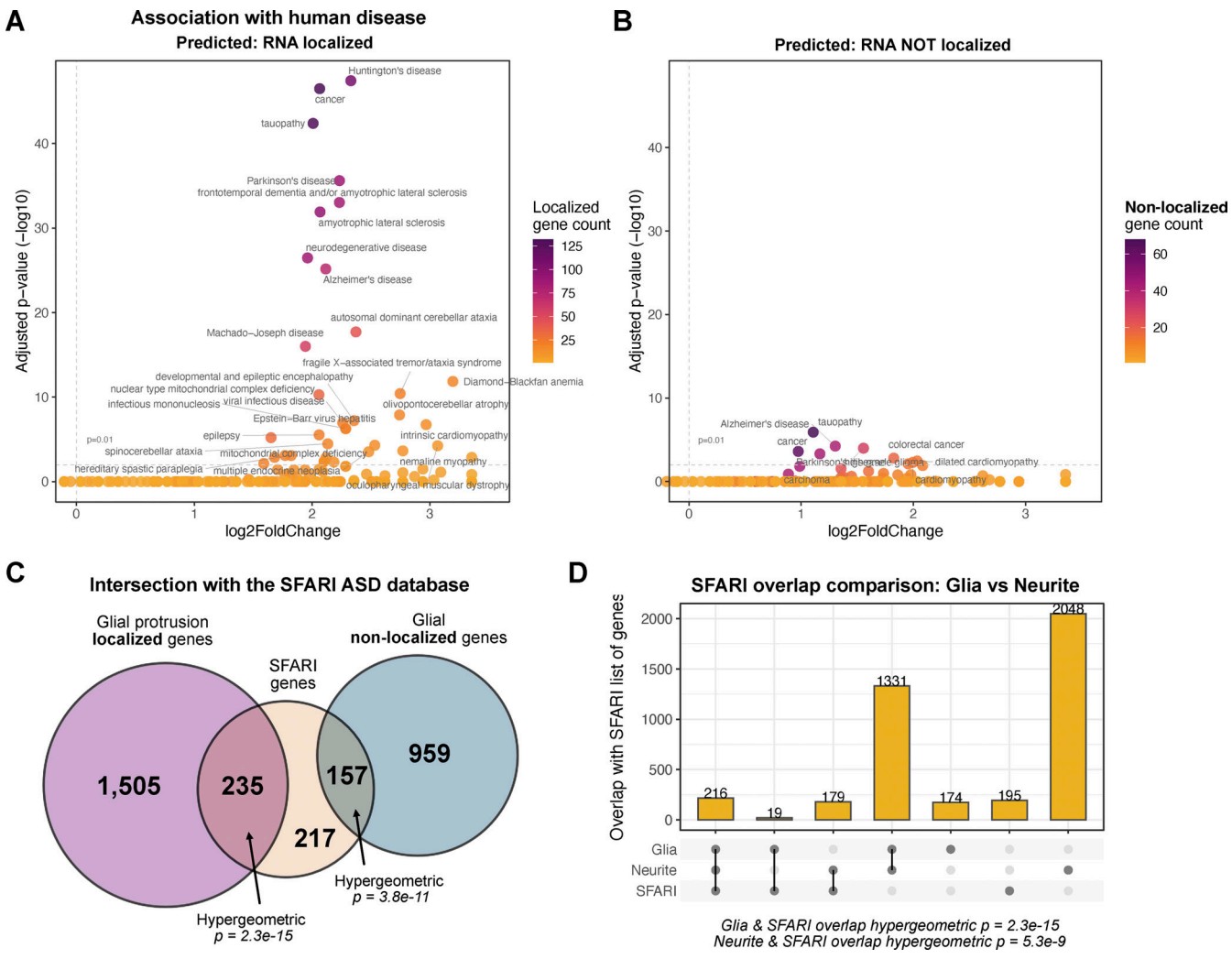

Figure 4. **Association of glial protrusion-localized transcripts with human disease. (A and B)** Enrichment of disease ontology (DO) terms in (A) the 1,740 glial protrusion-localized transcripts or (B) predicted to be not localized in glial protrusions. DO terms from *D. melanogaster* genes with high-confidence orthologs to *M. musculus* genes (DIOPT score ≥8) were used as background. Enrichment was assessed with a hypergeometric test and corrected for multiple hypothesis testing. **(C)** Overlap between glial protrusion-localized or non-localized transcripts with the SFARI list of genes. SFARI gene is an annotated list of genes that have been investigated in the context of autism spectrum disorder (Banerjee-Basu and Packer, 2010). 1,095 *H. sapiens* SFARI genes were converted to *D. melanogaster* genes (DIOPT score ≥8) before the statistical enrichment. **(D)** Comparison of overlap between glial localized protrusion transcripts and neurite-localized transcripts with the SFARI list of genes.

## More than 70% of the previously identified glial transcripts are predicted to be present in the *Drosophila* PNS glia

Do the predicted transcripts indeed localize to the periphery of glial cells? To test our prediction in *Drosophila* glia, we cross-referenced the 1,740 transcripts against a list of 19 transcripts, which we previously identified experimentally to be present in the protrusions of *Drosophila* peripheral glia, by screening 200 randomly chosen genes for their localization status throughout the nervous system, including NMJ glia (Titlow et al., 2023). We found that 15 out of 19 fly genes had high-confidence mammalian homologs, so we carried out our comparison only with these 15 genes (see Table 2).

We found that 11 out of the 15 transcripts (73.33%) were predicted from our analysis of mammalian glial transcripts to localize to the glial periphery in *Drosophila*. As these data originate from a systematic study of 200 mRNAs across the larval nervous system (Titlow et al., 2023), we sought to repeat the

experiments testing the localization status of the 11 genes in greater detail using smFISH together with markers of specific glial sub-types (Fig. 5 and Fig. S3; see Experimental methods for details). To this end, we visualized localized transcripts of venusYFP-containing mRNA in the extended NMJ glial protrusions of the YFP insertion lines in each gene (Lowe et al., 2014). The smFISH probe set against venusYFP was validated by showing the lack of fluorescent signal in untagged wild-type equivalent flies (Fig. S4). This result is consistent with our previous validation of the specificity and sensitivity of the YFP probe (Titlow et al., 2023). Our detailed smFISH experiments showed with greater confidence that all 11 transcripts were indeed present in the peripheral glia at the *Drosophila* NMJ. To determine the statistical significance of the 11/15 hits, we carried out an enrichment test that follows a hypergeometric distribution (independent sampling). Our results showed a slim probability

Table 2. **Table indicating whether the 19 genes previously predicted to be present in the *Drosophila* glia were present in the list of 1,740 *Drosophila* homologs**

| Found in Titlow et al | FBgn ID | Is predicted in 1,740? |
|---|---|---|
| *Lac* | FBgn0010238 | Yes |
| *Pdi* | FBgn0286818 | Yes |
| *nrv2* | FBgn0015777 | Yes |
| *Lost* | FBgn0263594 | No |
| *Flo-2* | FBgn0264078 | Yes |
| *alpha-Cat* | FBgn0010215 | Yes |
| *Vha55* | FBgn0005671 | Yes |
| *Atpalpha* | FBgn0002921 | Yes |
| *Nrg* | FBgn0264975 | Yes |
| *Cip4* | FBgn0035533 | Yes |
| *Nrx-IV* | FBgn0013997 | No |
| *gs2* | FBgn0001145 | Yes |
| *shot* | FBgn0013733 | Yes |
| *kst* | FBgn0004167 | No |
| *sdk* | FBgn0021764 | No |

Only the transcripts with high confidence ortholog pairs between mouse and fly were considered for the analysis (DIOPT score ≥8).

of $8.37 \times 10^{-05}$ for the 73% overlap to have occurred by chance, hence assuming a significant over-representation. We conclude that the predicted group of 1,740 is highly likely to be peripherally localized in *Drosophila* cytoplasmic glial projections.

Our secondary analysis also allowed us to obtain higher resolution single mRNA molecule information on the distributions of mRNAs molecules across the motoneuron axons and the surrounding glial protrusions (Fig. 5 and Fig. S3). We found specific examples of transcripts that were highly enriched in glia, with a much lower abundance in the motoneurons, including *gs2*, *nrv2*, *flo-2*, and *Lac* (Fig. 5 and Fig. S3, A–C). Other transcripts were uniformly distributed in motoneurons, glia, and muscle cells in the NMJ, including, *Pdi*, *shot*, *Vha55*, *Nrg*, *Cip4*, *alpha-Cat*, and *Atpalpha* (Fig. 5 F and Fig. S3, D–I). Interestingly, although *Pdi* mRNA is present in both glia and muscle, Pdi::YFP protein is only abundant in the glia (Fig. 5, F–I), indicating that translation of *Pdi* mRNA to protein is regulated in a cell-specific manner. We found the RNA localization of these 11 transcripts was supported by most mammalian libraries (Fig. S3 J), suggesting evolutionary conservation of RNA localization. In conclusion, these results show that the localization of transcripts, extrapolated from vertebrates to *Drosophila*, were highly concordant, reliably predicting the presence of multiple mRNAs at the glial periphery and highlighting a possible evolutionary conservation of glial protrusion-localized transcriptome.

## Localized glial transcripts influence plasticity in the adjacent motoneurons

To test if glial localized mRNAs influenced synaptic plasticity in the neighboring motoneurons, we knocked down the genes

specifically in glia, with the motoneurons remaining unaffected. We used UAS-RNAi lines against 11 different transcripts driven by the pan-glial *Repo-GAL4* driver, which is active in all three glial subtypes that make distal cell projection to NMJ motoneuron neurites (Fig. 1 and Fig. S1, A–C). We assayed the impact of knocking down each gene in glia on the ability of the wild-type NMJ motoneurons to make new synapses in response to chemical stimulation of the nerve. We applied the well-established protocol for activity-dependent synaptic plasticity through spaced pulses of potassium applied to a living NMJ preparation (Ataman et al., 2008; Piccioli and Littleton, 2014; Roche et al., 2002; Vasin et al., 2014). We found that 10 of the UAS-RNAi lines were viable until the third instar larvae without causing excessive lethality. RNAi knockdown of *Cip4* in glia did not yield viable third-instar larvae, so this gene was excluded from further analysis. Therefore, we tested 10 out of 11 localized transcripts in glia for their involvement in regulating the synaptic plasticity of motoneurons. Independently, we validated the glial-specific knockdown of transcripts using smFISH probes targeting the endogenous transcripts, which showed a strong reduction in RNA signal in *Repo>RNAi* flies (Fig. S5). Interestingly, we found that Repo>*Vha55*-RNAi larvae display a small central brain phenotype (Fig. S6); however, the larvae were otherwise indistinguishable in size from their control counterparts, so they were assayed as with other genotypes.

We performed a well-established method to induce structural plasticity, based on spaced potassium stimulation assay to quantify the effect of glial-specific RNAi on motor neuron plasticity (Fig. 6 A) (Ataman et al., 2008; Fernandes et al., 2023; Piccioli and Littleton, 2014). For this, we counted the number of newly formed neuronal boutons defined as "ghost" boutons, which are immature structures that lack post-synaptic density markers such as Discs large 1 (Dlg1) protein (Ataman et al., 2008; Roche et al., 2002). We found that for 3 of the 10 transcripts (*Lac*, *gs2*, *Pdi*), the RNAi knockdown in *Drosophila* glia causes a reduction in the number of ghost boutons formed by motoneurons in response to pulses of high potassium (Fig. 6, B and C). We conclude that *Lac*, *gs2*, and *Pdi* are required within glia for the correct plasticity of the motor neuron synapses.

We also assayed the effect of knocking down the localized transcripts in the glia on the morphology of the glial projections and the wild-type axon terminals in the NMJ. We examined the morphologies of third instar glia expressing RNAi and wild-type motoneurons (Fig. 6, D and E) with or without potassium stimulation (Fig. 6, F and G). We found that, even in the absence of stimulation, the knock-down of many of the genes resulted in aberrant growth or shrinkage of glial projections as well as a change in the ratio of glial to neuron surface areas, suggesting a developmental defect. For example, RNAi against shot resulted in near elimination of the glial projections. While in the control larvae, the glial projections extend on the surface of the muscle cells, in shot-RNAi larvae the non-synaptic motor axon branches are covered with glia, but the glia lack any projections onto the surface of the muscle (Fig. 6, D and E). We also saw varying defects in the sizes of both glial and neuronal projections after spaced potassium stimulation (Fig. 6, F and G). Our results demonstrate that many of the localized glial transcripts are

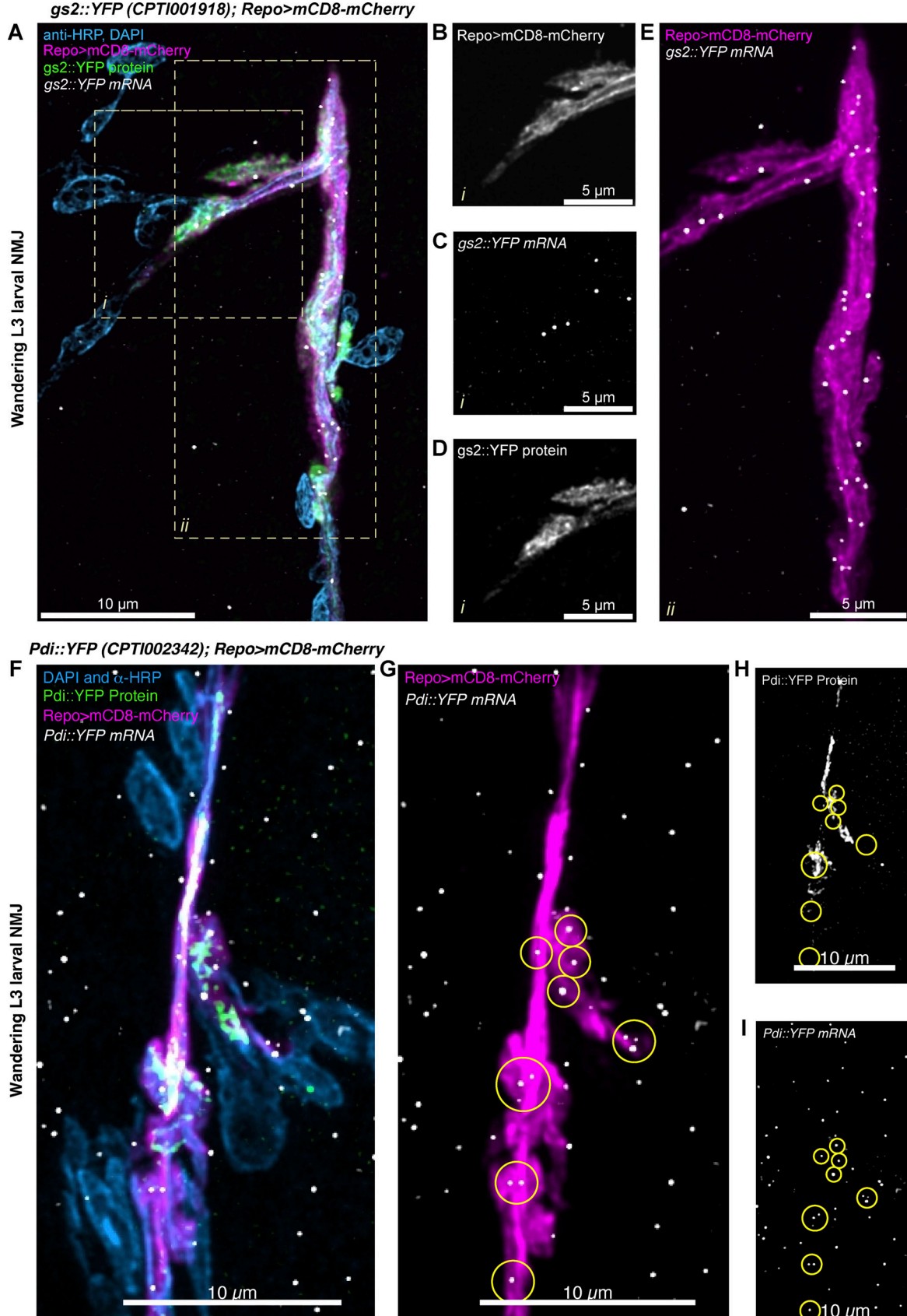

Figure 5.   **Localization of *gs2::YFP* and *pdi::YFP* transcripts in *Drosophila* glia. (A)** Representative image of a *Drosophila* third instar larva NMJ (Segment A4), showing the neuron and muscle cell nuclei in cyan (anti-HRP antibody conjugated to Alexa-405 fluor and DAPI, respectively), *gs2*::YFP protein in green, glial

membrane of perisynaptic glia labeled with Repo>mCD8-mCherry in magenta, and the YFP exon associated with *gs2*::YFP protein in white (ATTO633). **(B–D)** A single-channel view (see label) of a zoomed-in area (i) of (A) in which multiple mRNA molecules are present in the glial projection. **(E)** A zoomed in area (ii) of (A) showing the entire length of the perisynaptic glia at that synapse with the mRNA molecules distributed regularly throughout the glial cell projections. **(F–I)** Confocal image of third instar larval NMJ (Segment A4) of *Pdi*::YFP genotype. The same color scheme as (A) is used. Circular regions of interest in G–I represent individual clusters of *Pdi*::YFP mRNA localizing to the glial protrusion membrane.

required for the correct morphology of the *Drosophila* NMJ, and 30% are specifically necessary in glia for the correct synaptic plasticity of their neighboring neuron.

## Discussion

### Transcripts localized to the projections of different glial subtypes have conserved roles in coordinating synaptic plasticity and development

Here, we performed a meta-analysis of multiple published datasets of peripherally localized mammalian glial mRNAs to identify the core glial protrusion transcriptome shared across multiple glial subtypes. Our comprehensive analysis corresponds well with the previously published meta-analysis of neurite transcriptomes (von Kügelgen and Chekulaeva, 2020) and provides a valuable resource for future studies on glial mRNA localization. Using our dataset, we predicted 1,740 localized transcripts in three glial subtypes associated with the *Drosophila* motoneurons and NMJ synapses. These mRNAs are highly enriched in *Drosophila* homologs of human genes with associations in a variety of neurodegenerative and neuropsychiatric diseases, which also often coincide with molecular and cellular functions that have known associations with the cell periphery (Fig. 1). The localized glial mRNA that we predict include regulators of cytoskeleton dynamics and remodeling, membrane dynamics, signaling pathways, and mRNA metabolism (Fig. 3).

We validated our predictions by using smFISH to characterize in more detail the localization of 15 out of 19 of the mRNAs that we had previously identified as present in *Drosophila* glia (Titlow et al., 2023) and having high-confidence mammalian homologs. We found that 11 out of the 15 predicted transcripts were indeed localized in the glial periphery near the NMJ, suggesting that our meta-analysis holds strong predictive value, given that the probability of this happening by chance is ∼$10^{-5}$. Moreover, we found that for *Lac*, *GS2,* and *Pdi*, knocking down their expression by RNAi specifically in glia, while RNAi was not driven in motoneurons, the genes are specifically required for correct synaptic plasticity in the adjacent (wildtype) motoneurons. Our results do not explicitly test whether the localization of the mRNA in the glial periphery is specifically required for the genes to function in the glia and to influence plasticity in the adjacent motoneurons. Showing that would require future experiments to map the localization signals and then knock out localization in a conditional manner so that the mRNAs are only translated in the cell body and not in the cytoplasmic extensions of the glia. Future follow-up experiments could also include SunTag (Tanenbaum et al., 2014) and FLARIM (Richer et al., 2021, *Preprint*) to visualize local translation directly.

Knock-down of glial localized transcripts also causes morphological defects in the synapse (Fig. 6). Interestingly, morphological effects were not always associated with defects in activity-dependent plasticity and were also not always cell autonomous. These results highlight the complexity of intercellular communication necessary for proper synapse development and function. We conclude that our group of 1,740 mRNAs predicted to localize in *Drosophila* glia adjacent to the NMJ are likely to represent a rich resource of functionally relevant transcripts, which in many cases have associations with neurodegenerative, neuropsychiatric, and other important nervous system diseases, further emphasizing the power of data interoperability in biological research. Whilst we used the GAL4–UAS–RNAi system to knockdown target transcripts and validated the knockdowns using smFISH, we cannot completely exclude potential off-target effects of the RNAi system, and future studies should consider orthogonal mutant genotypes for detailed mechanical analysis.

It is important to consider whether the 1,740 transcripts that we predict are localized in the three glial subtypes associated with the NMJ are in fact a core localized glial transcriptome that is functionally active in the protrusions of any glial cell type. Certainly, the glial subtypes we characterized in *Drosophila* at the NMJ are also present, as defined by specific cell type markers, in association with neurons and synapses in the central brain. Furthermore, recent studies suggest that mRNA localization in one cell type mediated by interactions with RBPs can be predictive of the same interactions and localizations in other cell types with very different morphologies and functions (Goering et al., 2022, *Preprint*). It has also been shown that radial glia, astrocytes, and neurons, all of which have quite distinct morphologies and functions, share a significant overlap in the types of mRNAs localized to their protrusions (D'Arcy and Silver, 2020). Perhaps cytoplasmic peripheries of any kind of cell type share many universal properties in common, such as membrane trafficking and cytoskeletal dynamics.

### Glial protrusion transcriptomes contain a significant enrichment of disease ontology terms related to neuropathologies

Our analysis has indicated a significant enrichment of terms related to associations with nervous system diseases among the 1,740 *Drosophila* homologs with predicted glial mRNA localization (Fig. 4). Moreover, this group of genes has a statistically significant overlap with the SFARI database, which contains a list of genes implicated in Autism Spectrum Disorder (ASD). There is a growing interest in the role that glial cells may play in the mechanistic causes of diverse diseases related to nervous system development and function. Indeed, it has been suggested that mRNA localization and local translation could cause glial-induced pathological effects (Prater et al., 2022; Sloan and Barres, 2013). Our results bolster this idea, and the dearth of

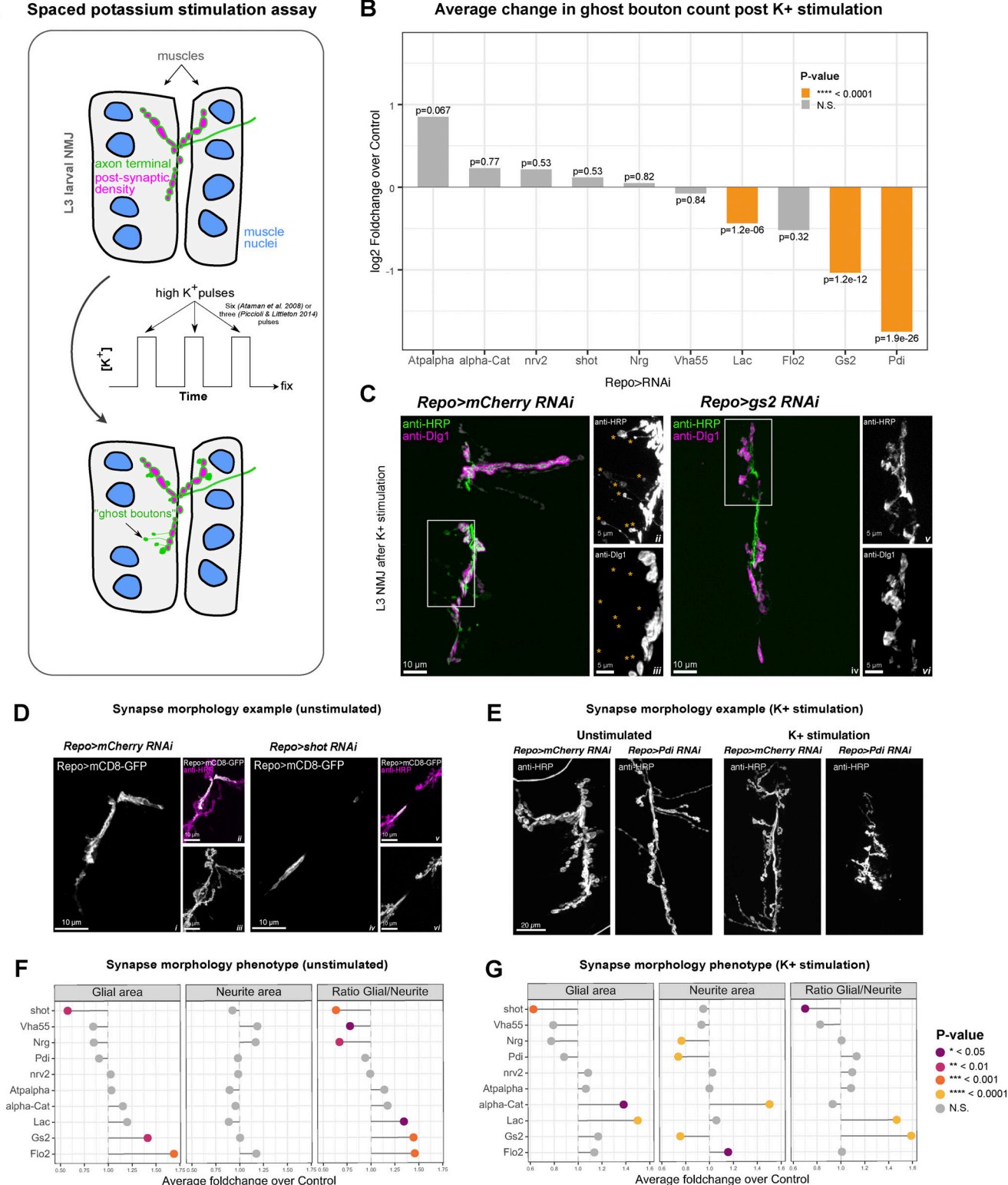

**Figure 6. Knockdown of glial localized transcripts interferes with synaptic plasticity and development. (A)** Schematic representation of the spaced potassium stimulation assay used in this study. Larvae are dissected and subjected to pulses of high potassium solution mimicking neuromuscular stimulation. After the assay has been completed, newly formed axon terminal endings with immature synapses, called "ghost boutons," can be detected using anti-HRP antibody labeling thanks to the lack of the post-synaptic density present, which can be labeled using anti-Discs-large antibody, which the ghost boutons lack. **(B)** Knock-downs of multiple glial protrusion-localized transcripts disrupt synaptic plasticity. Bar graph represents the average log$_2$ FoldChange of bouton counts post potassium stimulation compared to RNAi controls. Statistically significant changes are highlighted in orange color. Wilcoxon rank sum test P values are reported. N = 30 (*alpha-Cat*-RNAi), 30 (*Atpalpha*-RNAi), 58 (*Flo-2*-RNAi), 28 (*Vha55*-RNAi), 98 (*gs2*-RNAi), 238 (*Lac*-RNAi), 48 (*Nrg*-RNAi), 73 (*nrv2*-RNAi), 114 (*Pdi*-RNAi), 76 (*shot*-RNAi) NMJs. **(C)** Confocal images showing representative control (i–iii) and *gs2*-RNAi (iv–vi) synapses from the spaced potassium assay

experiment. Axon terminal and ghost boutons are labeled in green, and the anti-Discs-large antibody labeling in magenta in i and iv. ii and iii represent the area inside the white rectangle in i (see in-image labels for details). v and vi represent the area inside the white rectangle in iv. ii and iii show multiple ghost boutons lacking the post-synaptic density for the Control panel indicated with orange asterisks. v and vi show an extreme case where no ghost boutons were found for that NMJ is presented for the gs2-RNAi panel. **(D)** Confocal images showing representative control (i–iii) and shot-RNAi (iv–vi) synapses (see in-image labels for details). NMJ glial projections for shot-RNAi (iv) are much less expansive than the control in the unstimulated NMJs (i). **(E)** Confocal images showing representative control (i) and Pdi-RNAi (ii) synapses in unstimulated or K+ stimulated samples (see in-image labels for details). The NMJ areas are much smaller for the Pdi-RNAi NMJs when compared to the control in the stimulated NMJs. **(F and G)** Foldchange in glial protrusion area, neurite area, and their ratio upon knock-down of glial protrusion-localized transcripts before (F, N = 28 (alpha-Cat-RNAi), 30 (Atpalpha-RNAi), 30 (Flo-2-RNAi), 23 (Vha55-RNAi), 30 (gs2-RNAi), 28 (Lac-RNAi), 23 (Nrg-RNAi), 28 (nrv2-RNAi), 28 (Pdi-RNAi), 27 (shot-RNAi) NMJs) and after (G, N = 30 (alpha-Cat-RNAi), 30 (Atpalpha-RNAi), 58 (Flo-2-RNAi), 28 (Vha55-RNAi), 97 (gs2-RNAi), 239 (Lac-RNAi), 48 (Nrg-RNAi), 73 (nrv2-RNAi), 114 (Pdi-RNAi), 79 (shot-RNAi) NMJs) potassium activation assay. Data represents average foldchange for each gene. Student's t test.

literature in this area only serves to emphasize the need for more experimental work with a holistic approach to the nervous system, including glia and their communication with neurons at tripartite synapses. Although aggregated research made available in databases like SFARI mostly focuses on neuronal studies (Banerjee-Basu and Packer, 2010), our work shows that many SFARI genes are also transcribed into mRNA that is localized to glial protrusions. We have also shown that the reactome-pathway analysis of glial localized transcripts uncovers many enriched terms related both to signaling and mRNA metabolism, providing hints at potential unexplored mechanisms related to nervous system disorders. Another highly enriched reactome pathway term worth highlighting is the "innate immune system," which indicates that the availability of gene lists like ours is potentially valuable for multiple fields of research, as they extend beyond the context of the nervous system. Analysis modes like ours could prove useful in predicting which mRNAs might be localized in immune cells based on whether they are found in neurites.

An important direction of future research will be to characterize local translation at the peripheral cytoplasm of glia in response to signaling and neuronal activity, as was done in one recent study, which we included in our meta-analysis (Mazaré et al., 2020). The Drosophila larval system is a particularly good experimental system to study glia in relation to plasticity because of the ease of access to the NMJ and the ability to precisely assay the role of glia in the plasticity of the motoneurons. Moreover, the extensive genetic toolbox available in Drosophila makes the model very tractable and well-suited to studying the mechanistic effects of the introduction of disease mutations specifically into synaptic glia.

### Vertebrate localization data meta-analysis correctly predicts mRNA localization for the majority of transcripts

Using smFISH we confirmed, with greater precision, the presence of 11 transcripts of interest in Drosophila glial projections, which we had discovered in a recent survey of 200 transcripts across the larval nervous system (Titlow et al., 2023). We have focused on the glial cell subtypes that reach the Drosophila NMJ because of their extensive morphologies and close contact with the NMJ synapses (Fig. S1). The NMJ-associated glial subtypes are also present in the central brain, but unlike in the larval brain, the individual glial projections at the NMJ can be observed together with individual neuronal synapses. Building on our prior work (Titlow et al., 2023), we confirmed that the mammalian

localization data, which we reanalyzed and intersected with Drosophila glia, correctly predicts glial mRNA localization in Drosophila. We conclude that peripheral glial mRNA localization is a common and conserved phenomenon and propose that it is therefore likely to be functionally important. Naturally, our work is limited by the number of transcripts that could be tested experimentally. In the future, it will be important to continue to explore the localization of many more of the 1,740 Drosophila transcripts and test their functional requirement in glia as well as their potential implications for novel mechanisms of diseases of the nervous system.

## Materials and methods

### Meta-analysis of published transcriptomics datasets

We developed a custom RNA-seq analysis pipeline to process public transcriptomic datasets and allow the integration of diverse raw datasets from eight different studies. Raw FASTQ files were downloaded from Gene Expression Omnibus (GEO) and were filtered to remove ribosomal RNA reads using BBduk (Bushnell et al., 2017). Processed reads were mapped to Mus musculus (ENSEMBL release 96) or Rattus norvegicus (ENSEMBL release 98) genomes using Kallisto to obtain transcripts per million (TPM) abundance measures (Bray et al., 2016). A TPM value >10 for a transcript was considered a transcript to be present in a given compartment. To take into account intronic reads in libraries derived from soma, Kallisto bustools were used to create transcriptome indices containing intron sequences (Melsted et al., 2021). For the datasets where raw data were unavailable, the reported HTSeq count table was used as input, and the conflicting transcript annotations between ENSEMBL releases were manually resolved. In the case of non-Illumina-based sequencing datasets, numerical $\log_2$foldchange was calculated from the reported TPMs of each gene. R. norvegicus datasets were converted to M. musculus genes using ENSEMBL BioMart annotations after the differential expression processing. All summary data obtained from our analysis is available in Table S1.

Gene Ontology (GO) enrichment analyses were performed using the Bioconductor package clusterProfiler (Yu et al., 2012). Where applicable, multiple testing was corrected via the Bonferroni method. An adjusted P value < 0.01 and fold change > 1.5 was considered a significant enrichment. Enriched GO terms were further processed using the semsim() function implemented in the clusterProfiler package for simplification

based on semantic similarity. The full GO enrichment analysis table is provided in Table S2.

Conversion between human, mouse, and fly orthologs (ENSEMBL release 99) was performed using the source data of DIOPT version 9 (Hu et al., 2011). To select high-confidence orthologs, a cut-off score of 8 was used. Fly genes with the associated disease ontologies were acquired from FlyBase, and the enrichment of disease ontology terms was assessed with hypergeometric tests followed by Bonferroni's method of multiple hypothesis correction.

To identify transcripts that are expressed in the three glial cell types (perineurial, subperineurial, and ensheathing glial cells), single-cell nucleus sequencing data from Fly Cell Atlas was used (Li et al., 2022). The "Glial Cell" loom file from the 10x Cross-Tissue dataset was downloaded and filtered for non-glial cells based on the annotation. Using the R package ScopeLoomR (https://github.com/aertslab/SCopeLoomR), the digital gene expression matrices for the three cell types were extracted and the transcripts expressed in at least 5% of the corresponding clusters were identified.

For the Reactome pathway enrichment analysis, the gene-to-pathway query from Flymine was used to extract the Reactome ID annotations for each gene (https://www.flymine.org/flymine/templates/Gene_Pathway). The Reactome pathway hierarchical relationship between terms (available here) was used to group genes to the two highest hierarchical levels, the "top-level pathways" and the next hierarchical level, the "sub-pathways," which was used to calculate the enrichment.

### Experimental methods
#### Fly stocks
The fly lines used in this research were raised at 25°C (all stocks used in the smFISH experiments and the spaced potassium assay experiment apart from *Nrg*-RNAi) or 30°C (larvae used for the *Nrg*-RNAi spaced potassium assay experiment in glial cells) on a standard cornmeal-agar food. To label glial cells in the smFISH experiments, a cross was made to obtain the following stock: *UAS-mCD8-mCherry/CyoGFP*; *Repo>GAL4/Tm6B, Tb*. This stock was then crossed to each of the CPTI lines and offspring were selected for YFP and mCherry fluorescence. To label glia in the spaced potassium assay experiment with RNAi of transcripts of interest in glial cells, the following stock was constructed: *UAS-mCD8-GFP/UAS-mCD8-GFP*; *Repo>GAL4/Tm6B*, Tb. For each RNAi experiment, a homozygous line where a hairpin targets the coding sequence of the gene of interest with the least number of off-targets was selected. The UAS-RNAi lines were crossed with the pan-glial Repo-GAL4 driver (*UAS-mCD8-GFP/UAS-mCD8-GFP*; *Repo-GAL4/Tm6B,Tb*), and offsprings were selected for GFP fluorescence and the lack of *Tb* phenotype. The control for the UAS-RNAi experiment was *UAS-mCD8-GFP/+*; *Repo-GAL4/UAS-mCherry-RNAi*. We selected viable third-instar larvae for RNAi-based phenotypic assays and validated gene knockdowns by visualizing transcript expression in the NMJ glia. We did not recover any viable third instar larvae after glial-specific knockdown of *Cip4*, which was excluded from further analysis of synaptic plasticity phenotypes. We observed no adults of the

UAS-RNAi genotype for *nrv2*-RNAi, *Lac*-RNAi, *Cip4*-RNAi, and *Vha55*-RNAi. We observed adults for all other Repo>UAS-RNAi crosses. Table 3 lists all the strains used in the study.

### Solutions and reagents
The Hemolymph-Like Salines (HL3 solutions) were prepared as described previously (Ataman et al., 2008; Roche et al., 2002). The list of all the solutions used is given below (Table 4).

#### smFISH probes
Probes for the smFISH protocol were synthesized using a protocol described before (Gaspar et al., 2017). A set of oligonucleotides (28 for the YFP exon) against the gene region of interest was designed using LGC Biosearch Technologies' Stellaris RNA FISH Probe Designer. The oligonucleotides were pooled and elongated overnight at 37°C with a ddUTP conjugated to a desired dye (ATTO633 for the YFP probe) using terminal deoxynucleotidyl transferase enzyme from Life Technologies (Thermo Fisher Scientific). The fluorescently labeled oligonucleotides were then purified by Oligo Clean & Concentrator kit (Zymo Research) and eluted in TE buffer, after which their concentration and degree of labeling were measured using a NanoDrop spectrophotometer. The probes were diluted with a TE buffer to 25 μM concentration. All probe sequences used in this study are listed in Table 4.

### RNA single-molecule fluorescence in situ hybridization (smFISH) on the *Drosophila* larval fillet
RNA single-molecule fluorescence in situ hybridization (smFISH) was carried out as described previously (Titlow et al., 2018). In short, wandering third instar (L3) larvae were dissected in HL3 solution containing 0.3 mM $Ca^{2+}$ to produce a larval fillet with exposed NMJs, fixed for 30 min at room temperature (RT) using 4% paraformaldehyde in PBS containing 0/1% Triton-X (PBSTx), and permeabilized 2x for 20 min in PBSTx at RT. Samples were then prehybridized for 20 min at 37°C in wash buffer (2x SSC, 10% formamide [F9037; Sigma-Aldrich]) and then hybridized overnight at 37°C in hybridization buffer (10% formamide, 10% dextran sulfate [J62787.18; Alfa Aesar], 250 nM smFISH probe[s], and anti-HRP [Table 3] in 2x SSC). Samples were then rinsed in wash buffer again and counterstained with DAPI (1:1,000 from 0.5 mg/ml stock) in wash buffer for 45 min at RT. Next, samples were washed for 45 min in wash buffer at RT and incubated in Vectashield anti-fade mounting medium (Vector Laboratories) adjusted for the objective refractive index for 30 min and subsequently mounted on the slide in the mounting medium.

### Immunofluorescence (IF) on the *Drosophila* larval fillet
L3 larvae were dissected and fixed as described in the smFISH protocol. Larvae were then blocked for more than 1 h at 4°C in the blocking buffer (PBSTx, 1.0% BSA). Samples were incubated overnight at 4°C with the primary antibody (Table 5) in the blocking buffer. The next day, samples were washed for about 1 h and incubated in the secondary antibody solution (conjugated to Alexa Fluor 488, 568, or 647, used at 1:500, diluted in PBSTx; from Life Technologies)

Table 3. **List of *D. melanogaster* lines utilized in this study**

| Line name | Line ID | Line source | Line description |
|---|---|---|---|
| *nrv2*::YFP | CPTI001455 | Kyoto Stock Centre - Drosophila Genomics and Genetic Resources (DGRC) (https://www.dgrc.kit.ac.jp/) | *Nervana 2* gene YFP protein trap line (Lowe et al., 2014) |
| *Flo-2*::YFP | CPTI001427 | DGRC | *Flotillin 2* gene YFP protein trap line |
| *Cip4*::YFP | CPTI003231 | DGRC | *Cdc42-interacting protein 4* gene YFP protein trap line |
| *Vha55*::YFP | CPTI002645 | DGRC | *Vacuolar H+-ATPase 55kD subunit* gene YFP protein trap line |
| *Atpalpha*::YFP | CPTI002636 | DGRC | *Na pump α subunit* gene YFP protein trap line |
| *Nrg*::YFP | CPTI002761 | DGRC | *Neuroglian* gene YFP protein trap line |
| *Lac*::YFP | CPTI001714 | DGRC | *Lachesin* gene YFP protein trap line |
| *alpha-Cat*::YFP | CPTI002408 | DGRC | *α Catenin* gene YFP protein trap line |
| *Pdi*::YFP | CPTI002342 | DGRC | *Protein disulfide isomerase* gene YFP protein trap line |
| *gs2*::YFP | CPTI001918 | DGRC | *Glutamine synthetase 2* gene YFP protein trap line |
| Repo-GAL4 | RRID:BDSC_7415 | Bloomington Drosophila Stock Center (BDSC) (https://bdsc.indiana.edu/index.html) | Expresses GAL4 in glia |
| UAS-mCD8-mCherry/cyo-GFP | RRID:BDSC_27391 | BDSC | Expresses Cherry RFP fused to the mouse CD8 extracellular and transmembrane domains for membrane targeting under UAS control |
| UAS-mCD8-GFP/cyo-GFP | RRID:BDSC 63045 | BDSC | Expresses GFP fused to the mouse CD8 extracellular and transmembrane domains for membrane targeting under UAS control |
| Cyo-GFP/Gla; tm6, tb/pri | Crossed from BDSC stocks | BDSC | A double balanced stock carrying Cyo-GFP over gla on the 2nd chromosome and tm6, tb over prickle on the 3rd chromosome |
| *Atpalpha*-RNAi | RRID:BDSC 32913 | BDSC | Expresses dsRNA for RNAi of atpalpha under UAS control in the VALIUM20 vector |
| *alpha-Cat*-RNAi | KK107298 | VDRC | Expresses dsRNA for RNAi of *alpha-Cat* under UAS control in the pUAST vector pMF3 |
| *nrv2*-RNAi | GD960 | VDRC | Expresses dsRNA for RNAi of *nrv2* under UAS control in the pUAST vector pMF3 |
| *shot*-RNAi | RRID:BDSC 28336 | BDSC | Expresses dsRNA for RNAi of shot under UAS control in the VALIUM10 vector |
| *Vha55*-RNAi | GD 9363 | VDRC | Expresses dsRNA for RNAi of *Vha55* under UAS control in the pUAST vector pMF3 |
| *Lac*-RNAi | GD 35524 | VDRC | Expresses dsRNA for RNAi of *Lac* under UAS control in the pUAST vector pMF3 |
| *Flo-2*-RNAi | VSH 330316 | VDRC | Expresses dsRNA for RNAi of *Flo-2* under UAS control in the WALIUM20 vector |
| *gs2*-RNAi | GD 9378 | VDRC | Expresses dsRNA for RNAi of *gs2* under UAS control in the pUAST vector pMF3 |
| *Pdi*-RNAi | GD13418 | VDRC | Expresses dsRNA for RNAi of *Pdi* under UAS control in the pUAST vector pMF3 |
| *mCherry*-RNAi | RRID:BDSC 35785 | BDSC | Expresses dsRNA for RNAi of mCherry under UAS control in the VALIUM20 vector |
| *nrv2*-GAL4 | RRID: BDSC 6800 | BDSC | Expresses GAL4 in wrapping glia |
| 46F-GALl4 | – | Gift from Dr. Stefanie Schirmeier | Expresses GAL4 in perineurial glia (Xie and Auld, 2011) |
| *Mdr65*-GAL4 | RRID: BDSC 50472 | BDSC | Expresses GAL4 in subperineurial glia |

together with DAPI (1:1,000 from 0.5 mg/ml stock) for a further 1 h. Samples were then washed for 45 min in PBSTx at RT and incubated in Vectashield and mounted as in the smFISH protocol.

**Spaced High K⁺ depolarization paradigm**

The spaced potassium assay has been carried out as described before (Ataman et al., 2008; Piccioli and Littleton, 2014; Roche et al., 2002). Briefly, the larvae were dissected in 0.3 mM $Ca^{2+}$

**Table 4. List of all the solutions used in the project**

| Solution | Composition |
|---|---|
| HL3 0.3 mM Ca²⁺+ | NaCl 70 mM, KCl 5 mM, MgCl$_2$ 20 mM, NaHCO$_3$ 10 mM, trehalose 5 mM, HEPES 5 mM, sucrose 115 mM, Ca$^{2+}$ 0.3 mM, pH 7.2 |
| HL3 1 mM Ca²⁺ | NaCl 70 mM, KCl 5 mM, MgCl$_2$ 20 mM, NaHCO$_3$ 10 mM, trehalose 5 mM, HEPES 5 mM, sucrose 115 mM, Ca$^{2+}$ 1 mM, pH 7.2 |
| HL3 high K⁺ | NaCl 40 mM, KCl 90 mM, Ca$^{2+}$ 1.5 mM, MgCl$_2$ 20 mM, NaHCO$_3$ 10 mM, trehalose 5 mM, HEPES 5 mM, sucrose 5 mM, pH 7.2 |
| 10x PBS | 1.37 M NaCl, 27 mM KCl, 100 mM Na$_2$HPO$_4$, 20 mM KH$_2$PO$_4$, pH 7.4 |
| PBSTx | PBS 1x, 0.3% Triton X (vol/vol) |
| 20x SSC | 20 g NaCl, 100.5 g Tri-Sodium Citrate, pH 7.0 in 1 liter |
| TE | 10 mM Tris-HCL, 1 mM EDTA, pH 8.0 |

HL3 and then moved to 1 mM Ca²⁺ HL3. The unstretched larvae were then washed with high K⁺ HL3 for the periods of 2, 2, 2, 4, and 6 min with 15-min intervals of 1 mM Ca²⁺ HL3 in between (for RNAi experiments on *Atpalpha, alpha-Cat, nrv2, shot, Nrg, Flo-2, Vha55*) (Ataman et al., 2008). For RNAi knockdown experiments on *Lac, gs2, Pdi*, high potassium periods of 2 min, three times, with 10 min in between were used (Piccioli and Littleton, 2014). The assay was performed with a minimum of five control and five RNAi larvae per each experiment and in one replicate for the knockdown of *Atpalpha, alpha-Cat, Vha55*, two replicates for the knockdown of *Nrg* and *Flo-2*, six replicates for the knockdown of *Lac*, and three replicates for all remaining RNAi knockdown experiments. In each experiment, an internal control was performed at the same time, where the *Repo>mCD8-GFP* larvae were crossed to the UAS-RNAi line against mCherry protein, absent in these larvae, and each experiment was compared with its internal control only. After the spaced potassium pulses, the larvae were then stretched again, left for a period of rest of 30 min, and fixed and labeled as described in the IF

protocol. For all experiments, segments A2–A5 of muscles 6/7 were imaged.

### Image acquisition and processing

For smFISH experiments, the larvae were dissected, fixed, and stained with DAPI and anti-HRP antibody conjugated to the DyLight 405 dye, and the anti-YFP exon probe conjugated to ATTO633 dye, as per the smFISH protocol. A minimum of three larvae and 15 NMJs were assessed. For the potassium stimulation experiments, a minimum of five control and five RNAi larvae were dissected, treated as described above (see "Spaced High K⁺ depolarization paradigm"), fixed and stained with DAPI, anti-HRP antibody conjugated to Alexa-647 dye, and the anti-Dlg1 antibody as described in the IF protocol, and the glial membrane was labeled in endogenously with Repo>mCD8-GFP. Mounted specimens were imaged using an inverted Olympus FV3000 laser scanning confocal microscope or Olympus CSU-W1 SoRa laser spinning disk confocal microscope equipped with Prime BSI sCMOS camera using the Olympus Cellsens software. Images were acquired using 60×1.4NA Oil UPlanSApo objective (FV3000) or 100×1.45NA Oil UPlanSApo objective or 60× 1.5NA Oil UPlanSApo objective (SoRa). Laser units were solid state 405, 488, 568, and 640 lasers for both microscopes.

The images were then processed in ImageJ (https://imagej.nih.gov/ij/) (Schneider et al., 2012). The glial processes at the NMJ were largely flat, hence the two-dimensional areas of the GFP labeled glial membranes were measured as previously described; these membranes included small sections of non-synaptic motor axon branches from full Z-stack 2D projections as well as glial membrane nearing the synaptic lamella (Brink et al., 2012) The neurite areas were measured identically, except they were labeled with anti-HRP. The settings used were as follows: the linear auto-contrast, auto-threshold, and area measurement functions of NIH ImageJ (Abràmoff et al., 2004). In the spaced potassium stimulation experiment, the ghost boutons were quantified manually.

For statistical analysis, the R package *rstatix* was used. Each two-dimensional area from each NMJ at muscles constituted an

**Table 5. List of all the primary antibodies used in this study**

| Antibody | Detects | Source | Species | Cat no | Dilution |
|---|---|---|---|---|---|
| Anti-Dlg1 | Disks large, a post-synaptic scaffolding protein | Developmental Studies Hybridoma Bank, https://dshb.biology.uiowa.edu/ | Mouse | 4F3, RRID: AB_528203 | 1:200 |
| Anti-HRP, DyLight 405 | Neuronal membrane | Jackson ImmunoResearch Europe Ltd | Goat | 123-475-021, RRID: AB_2632561 | 1:100 |
| Anti-HRP, Alexa Fluor, 488 | Neuronal membrane | Jackson ImmunoResearch Europe Ltd | Goat | 123-545-021, RRID: AB_2338965 | 1:100 |
| Anti-HRP, Cy3 | Neuronal membrane | Jackson ImmunoResearch Europe Ltd | Goat | 123-165-021, RRID: AB_2338959 | 1:100 |
| Anti-HRP, Alexa Fluor 647 | Neuronal membrane | Jackson ImmunoResearch Europe Ltd | Goat | 123-605-021, RRID: AB_2338967 | 1:100 |

independent replicate ("*n*") value. The *n* numbers for each genotype and experiment are specified in respective figure legends. For the spaced potassium stimulation experiment, we quantified and reported the average $\log_2$ foldchange of bouton counts post-potassium stimulation compared with RNAi controls and performed the Wilcoxon rank sum test. For the areas' quantification, we calculated foldchange in glial protrusion area, neurite area, and their ratio upon knock-down of glial protrusion-localized transcripts. We performed these calculations in two sets: before and after the potassium activation assay. For each set, the average foldchange for each gene was calculated. Data normality was assessed using the Shapiro–Wilk test and Student's *t* tests were performed on the data to assess significant differences. The summary statistics are available in Table S3 (unstimulated NMJs areas, stimulated NMJs areas, and stimulated NMJs ghost boutons).

### Online supplemental material
Fig. S1 shows confocal images of *Drosophila* peripheral glia (related to Fig. 1). Fig. S2 shows functional annotation of glial protrusion localized and non-localized transcripts (related to Fig. 3). Fig. S3 shows localization of predicted transcripts in *Drosophila* glia (related to Fig. 5). Fig. S4 shows control experiment for venusYFP smFISH probe set. Fig. S5 shows validation of RNAi lines used in this study. Fig. S6 shows knockdown of Vha55 causes a small larval brain phenotype. Table S1 shows the summary table of glial protrusion transcriptome meta-analysis. Table S2 shows the Gene Ontology (GO) enrichment analysis of 1,740 glial protrusion localized genes output table. Table S3 shows summary data for neurite and glial area and potassium stimulation assay in RNAi experiments. Table S4 shows the smFISH probe sequences used in this study.

### Data availability
All analysis scripts and source data produced from this study are available at https://github.com/jefflee1103/Gala2023_glia-localised-RNA. The data that supports the findings of this study are available in the supplementary material of this article.

## Acknowledgments

We are very grateful to the Bloomington, Vienna, and Kyoto Drosophila Stock Centres (fly stocks) and Flybase for their reagents and open data, which were invaluable to this work. We are grateful to members of the Davis laboratory for critical reading of the manuscript and feedback on the project. We also thank Micron Oxford (https://micronoxford.com) for advanced microscopy facilities and technical advice.

This work was funded by a Wellcome Investigator Award 209412/Z/17/Z and Wellcome Strategic Awards (Micron Oxford) 091911/B/10/Z and 107457/Z/15/Z to I. Davis. M. Kiourlappou was supported by the Biotechnology and Biosciences Research Council (BBSRC), grant numbers BB/M011224/1 and BB/S507623/1. D.S. Gala is funded by Medical Sciences Graduate Studentships, University of Oxford. R.O. Teodoro is funded by PTDC-01778/2022–NeuroDev3D and iNOVA4Health–UIDB/04462/2020. Open Access funding provided by University of Oxford.

Author contributions: J.Y. Lee: Conceptualization, Data curation, Formal analysis, Methodology, Software, Validation, Visualization, Writing—review & editing, D.S. Gala: Conceptualization, Data curation, Formal analysis, Investigation, Methodology, Software, Supervision, Validation, Visualization, Writing—original draft, Writing—review & editing, M. Kiourlappou: Data curation, Formal analysis, Methodology, Software, Writing—review & editing, J. Olivares-Abril: Data curation, Formal analysis, Investigation, J. Joha: Investigation, Validation, J.S. Titlow: Conceptualization, Project administration, Supervision, Writing—review & editing, R.O. Teodoro: Methodology, Resources, Supervision, Writing—review & editing, I. Davis: Conceptualization, Funding acquisition, Project administration, Supervision, Writing—review & editing.

Disclosures: The authors declare no competing interests exist.

Submitted: 30 June 2023

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

# Supplemental material

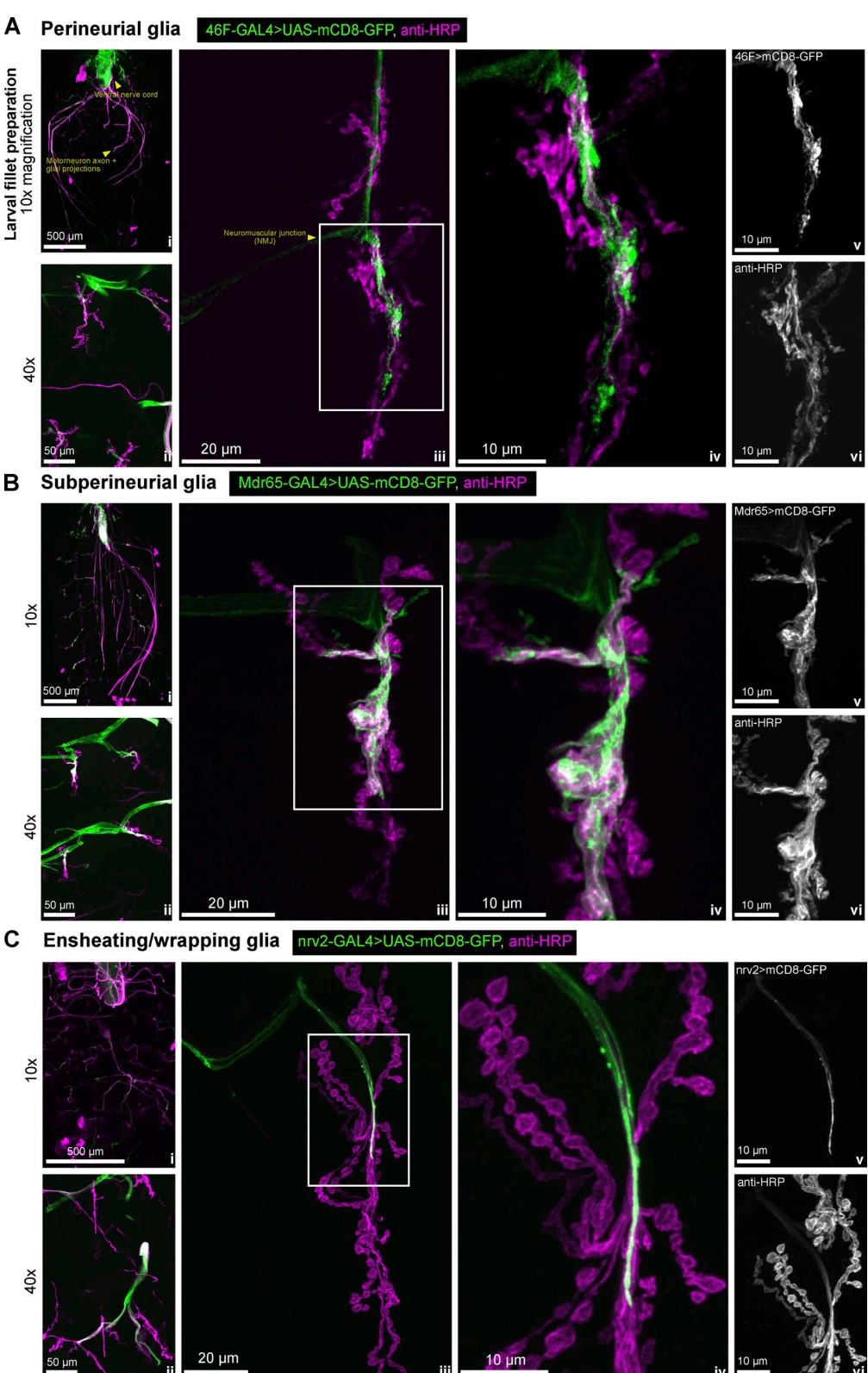

Figure S1.   **Confocal images of *Drosophila* peripheral glia (related to Fig. 1). (A–C)** Confocal images representing the location of (A) wrapping glia (*nrv2*-GAL4), (B) subperineurial glia (Mdr65-GAL4) and (C) perineurial glia (46F-GAL4) in the *Drosophila* third instar larva. All three of these glial subtypes produce extensive projections that reach hundreds of microns away from the cell body and reach the neuromuscular junction (NMJ) (middle two panels of each section). These three glial cell subtypes were chosen in our filtering for their elongated morphologies and the direct microscopically observable contact with the NMJ synapse. For each of A, B, and C: i represents the 10x overview of the whole dissected larva, ii represents a 40x zoom of segments A2 and A3 of muscles 6/7. iii presents an overview of the NMJ, and iv is a zoomed-in section of each of the images in iii, as indicated by the white boxes. v and vi show the individual channels from iv for clarity, with the channel label specified in the top left corner of v— glia and vi—anti-HRP.

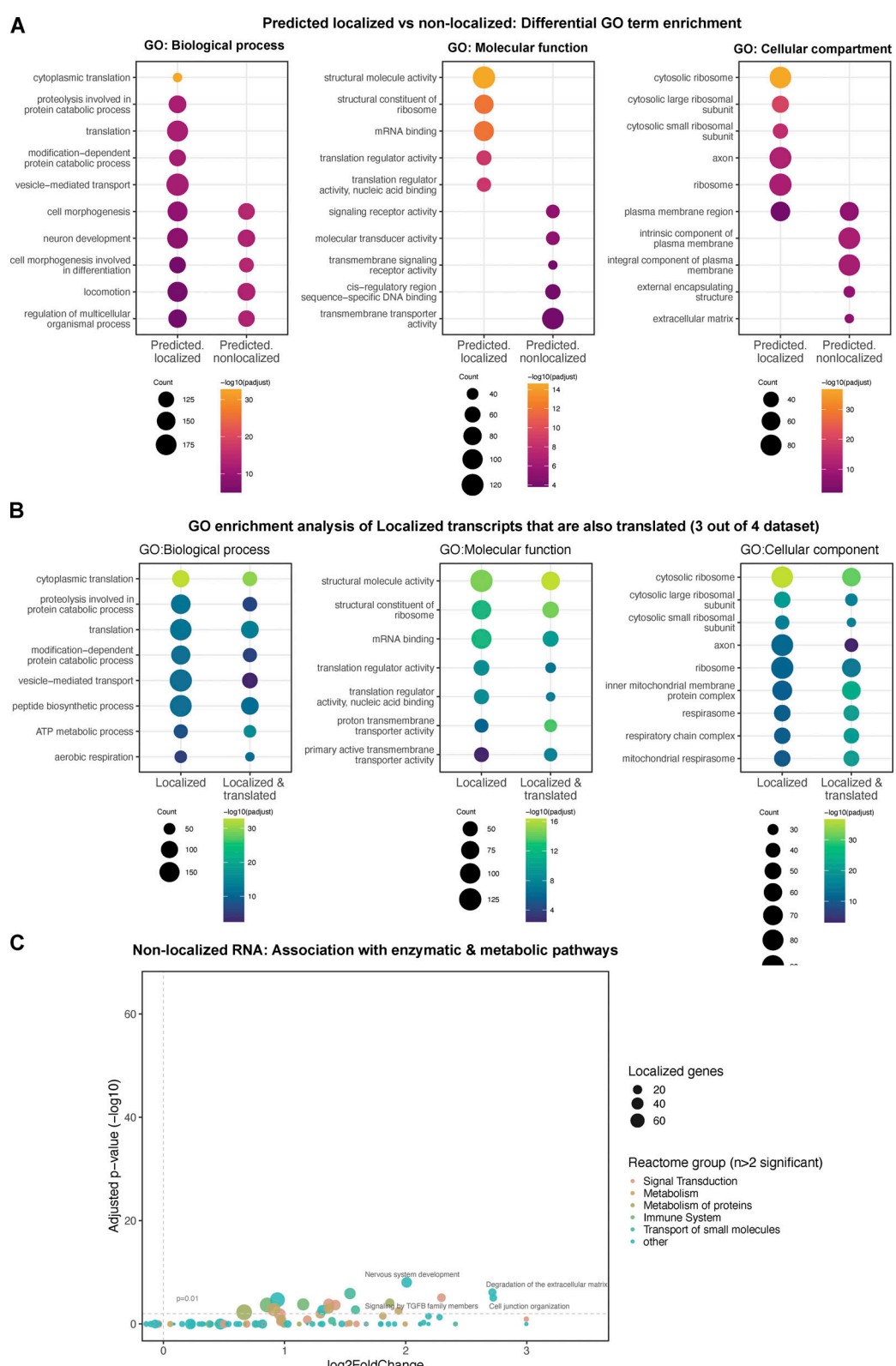

Figure S2. **Functional annotation of glial protrusion localized and non-localized transcripts (related to** Fig. 3**). (A)** Comparative Gene Ontology (GO) enrichment analysis between glial protrusion localized versus non-localized transcripts. The analysis was confined to transcripts expressed in perineurial, sub-perineurial, and ensheathing glial cells. *D. melanogaster* genes with high-confidence orthologs to *M. musculus* genes (DIOPT score ≥8) were used as background. The plots were generated using *enrichplot* R package, highlighting differentially enriched GO terms between the categories. **(B)** Comparative GO enrichment analysis between glial protrusion localized versus those that are also locally translated. Locally translated transcripts were identified from glial protrusion TRAP-seq libraries where TPM >10 and detectable in at least three out of the four libraries. **(C)** Reactome pathway enrichment analysis of the non-localized transcripts against the whole *D. melanogaster* transcriptome as background. The same analysis and plotting parameters were used as in Fig. 3 B.

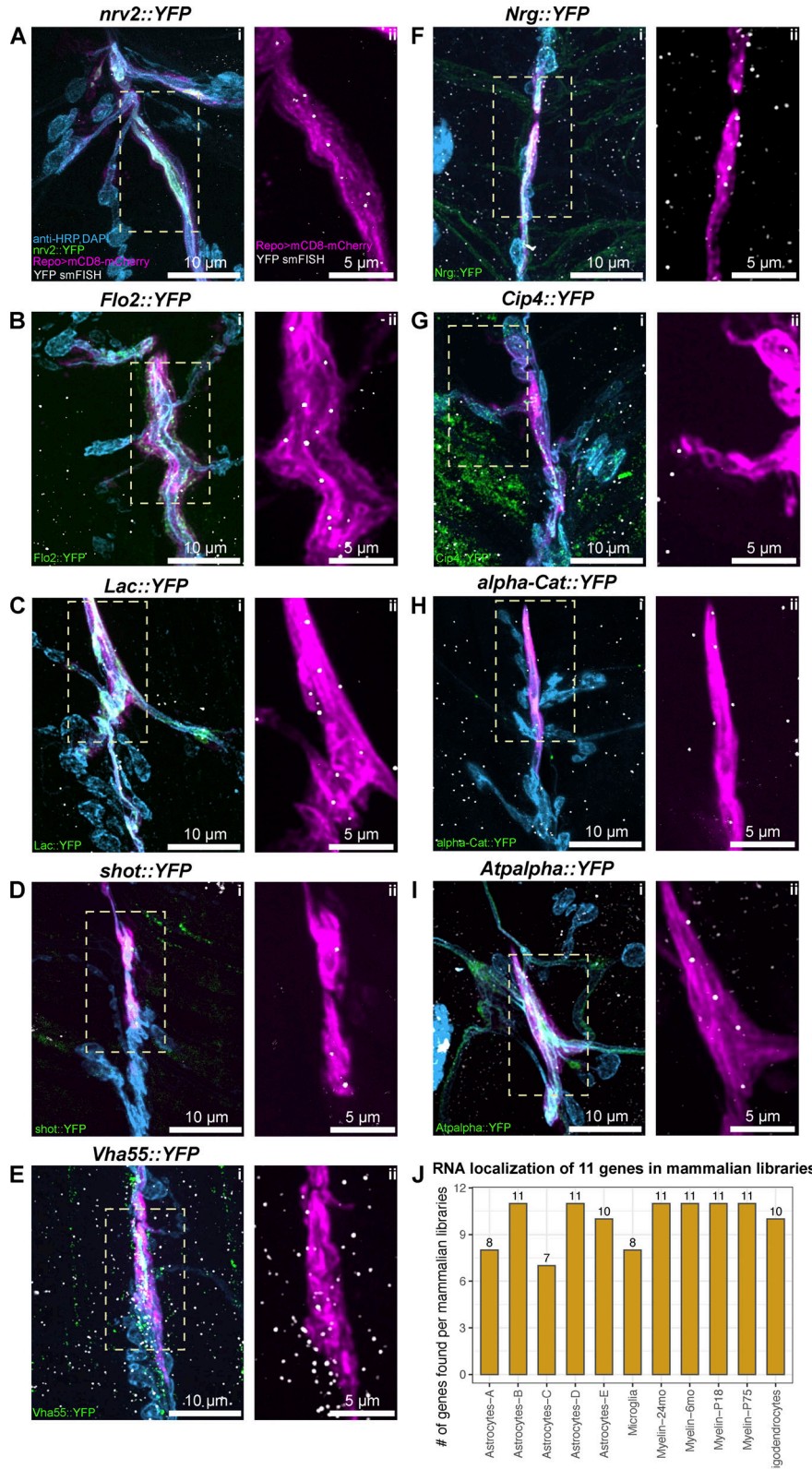

**Figure S3. Localization of predicted transcripts in *Drosophila* glia (related to** Fig. 5**). (A–I)** Confocal images of the *Drosophila* third instar larva NMJs (Segments A3 or A4), showing the (i) neuron and muscle cell nuclei in cyan (anti-HRP antibody conjugated to AlexaFluor405 and DAPI, respectively), YFP conjugated proteins in green (see in-image label for details), glial membrane of perisynaptic glia labeled with *Repo>mCD8-mCherry* in magenta, and the YFP exon associated with the protein in white (ATTO633). For all transcripts, mRNA was detected within glial membrane labeling, as indicated in ii for each panel, where a zoomed-in area denoted by a white rectangle in i is presented. **(J)** Bar plot showing how many of the 11 smFISH-validated transcripts were also detected in mammalian glial protrusion transcriptome datasets.

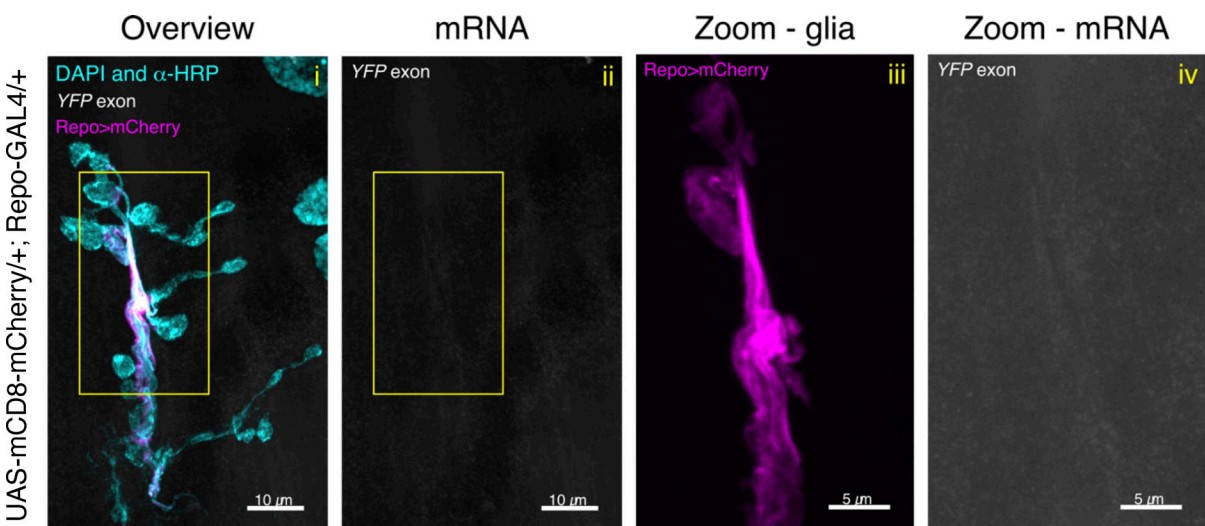

Figure S4. **Control experiment for venusYFP smFISH probe set.** Representative confocal microscopy image of a *Drosophila* third instar NMJ of wildtype control (*UAS-mCD8-mCherry/+*; *Repo-GAL4/+*). This fly line was constructed to label the glial cells (magenta, *Repo>mCherry*) and were hybridized with venusYFP smFISH probes (conjugated to ATTO633). DAPI and HRP antibody conjugated to AlexaFluor 405 fluorophore are depicted in cyan. The yellow ROI in the Overview (i) and mRNA (ii) panels (see top left corner for label details) is what is magnified in the Zoom-glia (iii, *Repo>mCherry* only) and Zoom-mRNA (iv, α-YFP probe only) panels. No specific fluorescent signal is observed in the smFISH channel.

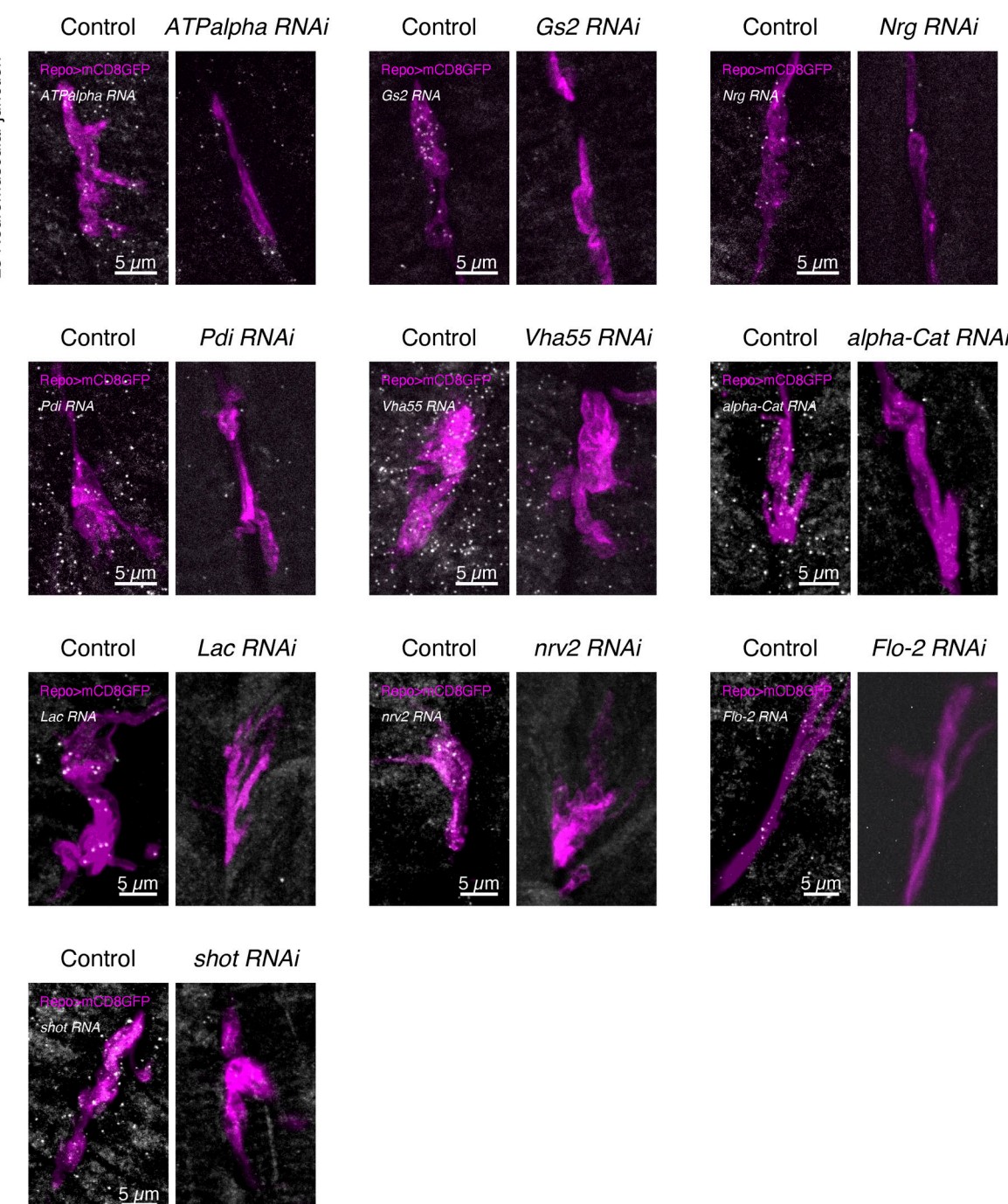

Figure S5. **Validation of RNAi lines used in this study.** Confocal images showing the reduced level of transcript expression using the *Repo-GAL4* driver in the third instar larval neuromuscular junction (NMJ) glial projections. smFISH probes against the endogenous genes were used to visualize transcripts in situ. The *Repo-GAL4* driver was used to drive both *UAS-RNAi* and *UAS-mCD8GFP* (magenta) to label glial protrusion membranes.

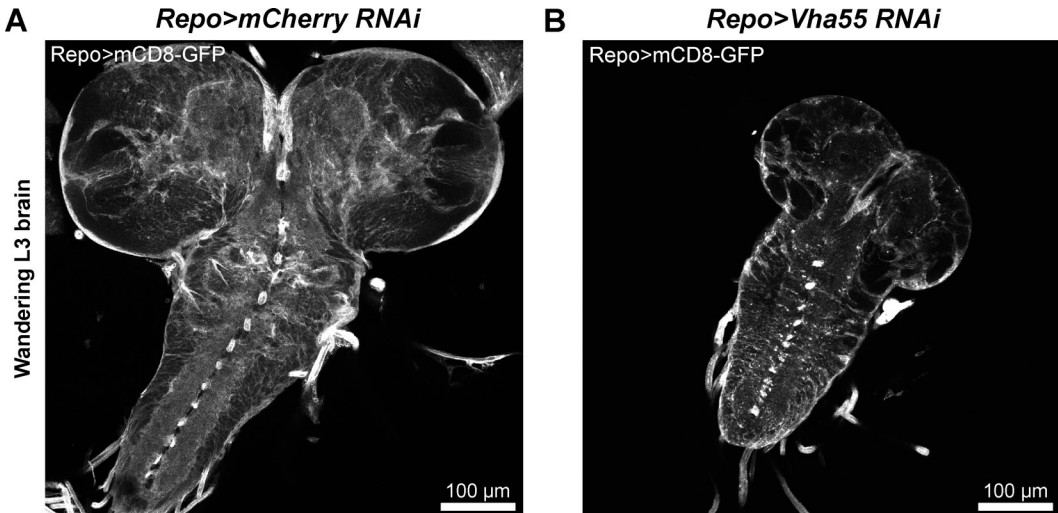

Figure S6. **Knockdown of *Vha55* causes a small larval brain phenotype. (A and B)** Confocal images showing the marked difference in size between the (A) control (*Repo-GAL4 > UAS-mCD8-GFP*; *UAS-mCherry-RNAi*) and (B) *Vha55*-RNAi (*Repo-GAL4 > UAS-mCD8-GFP*; *UAS-Vha55-RNAi*) brains for third instar larvae of very similar sizes (representative of five brains for each condition).

**Provided online are Table S1, Table S2, Table S3, and Table S4. Table S1 shows the summary table of glial protrusion transcriptome meta-analysis. Table S2 shows the Gene Ontology (GO) enrichment analysis of 1,740 glial protrusion localized genes output table. Table S3 shows summary data for neurite and glial area and potassium stimulation assay in RNAi experiments. Table S4 shows the smFISH probe sequences used in this study.**

