## [Peer Review File · The Journal of Cell Biology]

Murine glial protrusion transcripts predict localized *Drosophila* glial mRNAs involved in plasticity

Jeffrey Lee, Dalia Gala, Maria Kiourlappou, Julia Olivares-Abril, Jana Joha, Joshua Titlow, Rita Teodoro, and Ilan Davis

Corresponding Author(s): Ilan Davis, University of Glasgow

Review Timeline:

Submission Date:	2023-06-30
Editorial Decision:	2023-07-18
Revision Received:	2024-06-14
Editorial Decision:	2024-06-22
Revision Received:	2024-06-28

Monitoring Editor: Marc Freeman

Scientific Editor: Dan Simon

Transaction Report:

DOI: <https://doi.org/10.1083/jcb.202306152>

Revision 0

Review #1

1. Evidence, reproducibility and clarity:

Evidence, reproducibility and clarity (Required)

The study of mRNA localization in glia has been largely overlooked in comparison to neurons. However, glia are also polarized cells with long cytoplasmic processes that play important roles in neural function. In this manuscript, Gala and Lee et al attempt to fill in this missing void in the literature. They first use a meta-analysis of existing, cross-species, transcriptomic data to identify a set of 1,700 transcripts that are likely to be localized to the periphery of glia in *Drosophila*. They then used this list of transcripts to predict which mammalian glial transcripts are also likely to be localized. Their analysis suggests that a large proportion of the mammalian glial transcripts are predicted to be localized to the periphery of glia, and that these transcripts are enriched for functions involved in membrane trafficking, cytoskeleton regulation, local translation, and cell-cell communication. A connection with cross-cellular communication (ie glia-glia, glia-neuron, glia-muscle) at the neuromuscular junction, prompting the authors to assess if some of these transcripts play a role in plasticity of at the NMJ. Using siRNA driver lines, that loss of some of their assessed transcripts prevent new synapse formation- suggestive of a role for mRNA localization and local protein synthesis in driving synaptic plasticity. These findings suggest that mRNA localization is a widespread phenomenon in glia, and that it plays an equally important role as the more widely studied neuronal mRNA localization.

****Key Points****

- 1,700 transcripts are predicted to be localized to the periphery of PNS glia in *Drosophila* at the NMJ.
- A large proportion of the mammalian glial transcripts are predicted to be localized to the periphery of glia.
- Localized transcripts are enriched for functions involved in membrane trafficking, cytoskeleton regulation, local translation, and cell-cell communication.
- Localized glial transcripts may function in synaptic plasticity.

****Major comments****

Overall, the work presented within manuscript is well reasoned and supports the points the authors generally are trying to make. The work as is, does fall a little on the light side, resting on almost exclusive bioinformatic analysis with some limited validation. The presented work itself (Figures 1-4) does not need significant adjustments prior to publication, in my view, with only a few points to address. However, the work in Figure 5- doesn't really support the claims the authors make on its own, and would require some additional experiments or at the very least discussion of the caveats to its current form.

1. Localized glia transcripts, are they "glial/CNS/PNS" significant or are they similar to other known datasets of protrusion transcriptomes? The authors compared their 4801 "total" localized to a local transcriptome dataset from the Chekulaeva lab finding that a significant fraction are localized in both. As the authors note, this is in good agreement with a recent paper from the Talifarro lab showing conservation of localization of mRNAs across different cell types. What the authors haven't done here, is further test this by looking at other non-neuronal projection transcriptomic datasets (for example Mardakheh Developmental Cell 2015, among others). If the predicted glia-localized processes are similar to non-neuronal processes transcriptomes, this would further strengthen this claim and rule out some level of CNS/PNS derived lineage driving the similarities between glia and neuronal localized transcripts.
2. The presentation/discussion around Figure 3 is a bit weaker than other parts of the manuscript, and it doesn't really contribute to the story in its current form. Notably there is no discussion about the significance of glia in neurological disorders until the very end of the manuscript (page 21), meaning when its first brought up.. it just sits there as a one off side point. The authors might consider strengthening/tightening up the discussion here, if they really want to keep it as a solo main figure rather than integrating it somewhere else/putting it into supplemental. In my view, Figures 2 & 3 should be merged into something a bit more streamlined.
3. Why aren't there more examples of different mRNAs in Figure 4? Seems a waste to kick them all to supplemental.
4. The plasticity experiments, while creative, I think need to be approached far more cautiously in their interpretation. Given that the siRNAs will completely deplete these mRNAs- it really needs to be stressed any/all of the effects seen could just be the result of "defective" or "altered" states in this glial population- which has spill over effects on plasticity in at the NMJ. Without directly visualizing if these mRNAs are locally translated in these processes and assessing if their translation is modulated by their plasticity paradigm, all these experiments can say is that these RNAs are needed in glia to modulate ghost bouton formation in axons. This represents the weakest part of this manuscript, and the part that I feel does not actually backup the claims currently being made. Without any experiments to A. quantify how much of these transcripts are localized vs in the cell body of these glia, B. visualize/quantify the translation of these mRNAs during baseline and during plasticity; the authors cannot use these data to claim that localized mRNAs are required for synaptic plasticity.

****Minor points:****

1. The use of blue/green or blue/green/magenta is difficult to resolve in some places. Swapping blue for cyan would greatly aid in visualizing their data.
2. Make the colouring/formatting of the tables more consistent, its distracting when its constantly changing (also there is no need for a blue background.. just use a basic white table).

2. Significance:

Significance (Required)

This work would be well received within the field, being at a time where increasing focus on mRNA localization and local protein synthesis are major players in glia along with the more

widely studied neurons. Additionally, that mRNAs localized in glia are relevant to neurological disorders also comes at a time where increasingly, especially in neurodegenerative disorders, glia are believed to be drivers of the diseases as well- suggesting that dysregulation of local protein synthesis in glia may play a role. The major weakness of this work is that being largely bioinformatic (from existing datasets), a lot of this is speculative and no conclusive data is shown to demonstrate how/that this glia population is utilizing mRNA localization in protrusions to fuel local protein synthesis for a specific purpose (ie synaptic plasticity).

This work would be of interested to those broadly interested in glial biology and mRNA localization.

The reviewers background is in RNA localization and local protein synthesis in neurons.

3. How much time do you estimate the authors will need to complete the suggested revisions:

Estimated time to Complete Revisions (Required)

(Decision Recommendation)

Between 1 and 3 months

Yes

Review #2

1. Evidence, reproducibility and clarity:

Evidence, reproducibility and clarity (Required)

****Summary:****

The manuscript by Gala et al. describes efforts to systematically identify candidate localizing mRNAs within glial cells of the *Drosophila* larval nervous system. They address this issue with an interesting and novel approach that involves compiling candidate localized transcripts from available mammalian cell datasets and cross-referencing a list of mRNAs from homologous *Drosophila* genes with a small number of candidates for mRNA localization in *Drosophila* glia cells that this team identified in a previous study (which they have now validated further). The results provide compelling evidence that their pipeline has predictive power. The authors then go on to perform functional analysis of a small number of validated localizing transcripts and provide evidence that several of these genes are functionally important in glia, including in processes related to synaptic plasticity. These observations raise the possibility that the localization process is functionally important but this was not tested directly (doing so would require a separate long-term study).

The manuscript is well written and the figures are of high quality. Methodological details are provided, as is quantification where appropriate. However, some points are not yet sufficiently strongly supported by the data yet and the justification for some aspects of the workflow needs clarifying.

****Major points:****

1. The authors analyse the 1700 shortlisted genes for Gene Ontology and associations with autism spectrum disorder, leading to interesting results. However, it is not clear to what extent the enrichments they observe are driven by their presumptive localization or if the associations are driven to a significant extent by the presence of these genes in the selected cell types in the Fly Cell Atlas. One way to address this would be to perform the GO and SFARI analysis on genes that are expressed in the same cells in the Fly Cell Atlas but were not shortlisted from the mammalian cell datasets - the results could then be compared to those obtained with the 1700 localized transcripts.
2. Although the authors attempt to justify its inclusion, I'm not convinced why it was important to use the whole cell transcriptome of perisynaptic Schwann cells as part of the selection process for localizing transcripts. Including this dataset may reduce the power of the pipeline by including mRNAs that are not localized to protrusions. How many of the shortlisted 1700 genes, and how many of the 11 glial localized mRNAs in Table 5, would be lost if the whole cell transcriptome were excluded. More generally, what is the distribution of the 11 validated localizing transcripts in each dataset in Table 4? This information might be valuable for determining which dataset(s), if any, has the best predictive power in this context.
3. Did the authors check if any of the RNAi constructs are reducing levels of the target mRNA or protein? Doing so would strengthen the confidence in these important results significantly. In any case, the authors should also mention the caveat of potential off-target effects of RNAi.
4. Methods: what is the justification for assuming that if the RNAi cross caused embryonic or larval lethality then the 'next most suitable' RNAi line is reporting on a phenotype specific to the gene. If the authors want to claim the effect is associated with different degrees of knockdown they should show this experimentally. An alternative explanation is that the line used for phenotypic analysis in glia is associated with an off-target effect.

****Minor points:****

1. It would be helpful to have in the Introduction (rather than the Results, as is currently the case) an operational definition of mRNA localization in the context of the study. And is it known whether or not localization in protrusions is the norm in mammalian glia or the *Drosophila* larval glia? I ask because it may be that almost all mRNAs diffuse into the protrusion, so this is not a selective process. One interesting approach to test this idea might be to test if the 1700 shortlisted transcripts have a significant underrepresentation of 'housekeeping' functions.
2. Introduction: 'Asymmetric mRNA localization is likely to be as important in glia, as it is in neurons,...'. Remove commas

2. Significance:

Significance (Required)

Whilst glial mRNA localization has been reported in other systems, this study is significant as it paves the way for functional and mechanistic dissection of this process in a genetically tractable model system that involves physiologically relevant neuron-glia interactions. The successful results of the data mining approach may also encourage others to address other problems in this manner. The study would be more impactful if additional hits from the 1700 transcripts were validated for glial localization but the findings in the current manuscript are still important. The advance is more methodological than conceptual. The work will appeal to cell biology and systems biology audiences.

3. How much time do you estimate the authors will need to complete the suggested revisions:

Estimated time to Complete Revisions (Required)

(Decision Recommendation)

Between 1 and 3 months

4. *Review Commons* values the work of reviewers and encourages them to get credit for their work. Select 'Yes' below to register your reviewing activity at Web of Science Reviewer Recognition Service (formerly Publons); note that

the content of your review will not be visible on Web of Science.

No

Review #3

1. Evidence, reproducibility and clarity:

Evidence, reproducibility and clarity (Required)

In the manuscript by Gala et al, the authors perform meta-analysis of transcriptomic datasets from different studies focused on synaptically-associated mammalian glial cells. Based on a detailed analysis, the authors predict 1700 localized transcripts and attempted to identify the local transcriptome in glial cells conserved in *Drosophila* and mammals. This kind of data mining is informative, and can predict the top candidates relatively well, however, the study suffers from lack of depth and a careful assessment comparing across the different datasets. The main concerns have been listed below and suggestions for a more detailed and careful assessment of the data.

1. The authors have pooled data from different studies across different type of glial cells performed from in vitro to in vivo. While pooling datasets may reveal common transcripts enriched in processes, this may not be the best approach considering these are completely different types of glial cells with distinct function in neuronal physiology.
2. It is important to note the limitations of the study. For example, DeSeq2 is biased for highly expressed transcripts. How robust was the prediction for low abundance transcripts?
3. The authors identify 1,700 transcripts that they classify as "predicted to be present" in the projections of the *Drosophila* PNS glia. This was based on the comparison to all the mammalian glial transcripts. Since the authors have access to a transcriptomic study from Perisynaptic Schwann cells (PSCs), the nonmyelinating glia associated with the NMJ isolated from mice; it would be more convincing to then validate the extent of overlap between *Drosophila* peripheral glial with the mammalian PSCs. This may reveal conserved features of localized transcripts in the PNS, particularly associated with the NMJ function.
4. Fig 2: What is the extent of overlap between the translating fractions versus the localized fraction? It will be informative to perform the functional annotation of the translating glial transcripts as identified from Fig 1D.
5. "We conclude predicted group of 1,700 are highly likely to be peripherally localized in *Drosophila* cytoplasmic glial projections". To validate their predictions, the authors test some of these candidates in only one glial cell type. It might be worthy to extend this for other differentially expressed genes localized in another glial type as well.
6. Figure 5: The authors perform KD of candidate transcripts to test the effect on synapse formation. However, these are KD with RNAi that spans across the entire cell. To make the

claim about the importance of "target" RNA localization in glia stronger, ideally, they should disrupt the enrichment specifically in the glial protusions and test the impact on bouton formation. Do these three RNAs have any putative localization elements?

7. RNA localization in oligodendrocytes has been well studied and characterized. The authors should cite and discuss those papers (PMID: 18442491; PMID: 9281585).

2. Significance:

Significance (Required)

In the manuscript by Gala et al, the authors perform meta-analysis of transcriptomic datasets from different studies focused on synaptically-associated mammalian glial cells. Based on a detailed analysis, the authors predict 1700 localized transcripts and attempted to identify the local transcriptome in glial cells conserved in Drosophila and mammals. This kind of data mining is informative, and can predict the top candidates relatively well, however, the study suffers from lack of depth and a careful assessment comparing across the different datasets and the biological significance.

3. How much time do you estimate the authors will need to complete the suggested revisions:

Estimated time to Complete Revisions (Required)

(Decision Recommendation)

Between 3 and 6 months

Yes

Review #4

1. Evidence, reproducibility and clarity:

Evidence, reproducibility and clarity (Required)

Peripheral localization of mRNAs to cellular protrusions (e.g. axons, astrocyte peri-synaptic processes, myelin sheaths) is important for activity-dependent regulation of nervous system function. In Gala et al., the authors aim to identify conserved, peripherally located mRNAs in through a combination of unbiased transcriptomics and in vivo validation in *Drosophila* NMJ glia. To that end, they mine several published datasets enriched for peripherally located mRNAs in glia, identify which of these transcripts are conserved in fly, filtered for mRNAs that are enriched in processes over soma, and filtered for mRNAs that were enriched in three glial subtypes that ensheath their model system: *Drosophila* NMJ. Finally, they go on to validate 11 (of 15) predicted genes as located in NMJ glia, demonstrating that loss of these transcripts, in some cases, impacts synaptic plasticity.

This is an interesting study that complements a growing interest in the community: how do glia locally support the neurons that interact with. I have a number of suggestions to support the conclusions made in this manuscript:

****Major:****

1. The authors use FISH to validate the glial expression of their target genes, though these experiments are not quantified, and no controls are shown. The authors should provide a supplemental figure with "no probe" controls, and/or validate the specificity of the probe via glial knockdown of the target gene (see point 2). Furthermore, these data should be quantified (e.g. number of puncta colocalized with NMJ glia membranes).
2. For the most part, the authors only use one RNAi line for their functional studies, and they only show data for one line, even if multiple were used. To rule out potential false negatives, the authors should leverage their FISH probes to show the efficacy of their knockdowns in glia. This would serve the dual purpose of validating the new probes (see point 1).
3. In Figure 5 E, given the severe reduction in size in the stimulated Pdi KD animals, the authors should show images of the unstimulated nerve as well. Do the nerve terminals actually shrink in size in these animals following stimulation, rather than expand? The NMJ looks substantially smaller than a normal L3 NMJ, though their quantification of neurite size in F suggests they're normal until stimulation.

****Minor:****

1. In Figure 5 D, the authors should include a label to indicate that these images are from an unstimulated condition.
2. The authors claim that there is an enrichment of ASD-related genes in their final list of ~1400 genes that are enriched in glial processes. It is well-appreciated that synaptically-localized mRNAs are generally linked to ASDs. Can the authors comment on whether the transcripts localized to glial processes are even more linked to ASDs and neurological disorders than transcripts known to be localized to neuronal processes?
3. The authors are missing a number of key citations for studies that have explored the functional significance of mRNA trafficking in glia, and those that have validated activity-dependent

translation:

- <https://pubmed.ncbi.nlm.nih.gov/18490510/>
- <https://pubmed.ncbi.nlm.nih.gov/7691830/>
- <https://journals.plos.org/plosbiology/article?id=10.1371/journal.pbio.3001053>
- <https://www.ncbi.nlm.nih.gov/pmc/articles/PMC7450274/>
- <https://pubmed.ncbi.nlm.nih.gov/36261025/>

2. Significance:

Significance (Required)

Glia are morphologically complex cells that extend tens (microglia/oligodendrocytes) to thousands (astrocytes) of processes to interact with and support neurons. Given this complex morphology, and the ability of glia to dynamically respond to changes in neuronal activity, there has been a push in recent years to characterize local mechanisms of glia-neuron support. These mechanisms include mRNA trafficking and activity-dependent translation in distal processes. The authors of this study did a nice job computationally identifying a set of putative genes that are enriched in glial processes, which should be of broad interest to the glial community.

3. How much time do you estimate the authors will need to complete the suggested revisions:

Estimated time to Complete Revisions (Required)

(Decision Recommendation)

Between 1 and 3 months

No

Revision Plan

Manuscript number: RC-2023-01938R

Corresponding author(s): Ilan, Davis

1. General Statements

We thank all four reviewers for their helpful and constructive comments. We have gone through each and every comment and proposed how we would address each point raised by the reviewers. We are confident our proposed revisions are feasible within a reasonable and expected time frame. Some of the comments regarding minor typo/aesthetics and extra references have already been addressed in the transferred manuscript. The changes are highlighted in yellow in the transferred manuscript.

2. Description of the planned revisions

Reviewer #1 -----

Major points:

1. The presented work itself (Figures 1-4) does not need significant adjustments prior to publication, in my view, with only a few points to address. However, the work in Figure 5- doesn't really support the claims the authors make on its own, and would require some additional experiments or at the very least discussion of the caveats to its current form.

We thank the reviewer for these comments and will follow the reviewer's suggestion by discussing the caveats regarding the interpretation of Figure 5. We will also add to the discussion to suggest future research approaches beyond the scope of this manuscript that would address the functional importance of localised mRNA translation. We will briefly mention in the discussion methods such as the quantification of the mRNA foci and the disruption of the mRNA localisation signals to disrupt localised translation and the use of techniques such as Sun-Tag (Tanenbaum et al, 2014) and FLARIM (Richer et al, 2021) to visualise local translation directly.

Tanenbaum et al, 2014 DOI: 10.1016/j.cell.2014.09.039

Richer et al, 2021 DOI: 10.1101/2021.08.13.456301

2. Localized glia transcripts, are they "glial/CNS/PNS" significant or are they similar to other known datasets of protrusion transcriptomes? The authors compared their 4801 "total" localized to a local transcriptome dataset from the Chekulaeva lab finding that a significant fraction are localized in both. As the authors note, this is in good agreement with a recent paper from the Talifarro lab showing conservation of localization of mRNAs across different cell types. What the authors haven't done here, is further test this by looking at other non-neuronal projection transcriptomic datasets (for example Mardakheh Developmental Cell 2015, among others). If the predicted glia-localized processes are similar to non-neuronal processes transcriptomes, this would further strengthen this claim and rule out some level of CNS/PNS derived lineage driving the similarities between glia and neuronal localized transcripts.

Revision Plan

This is a good point and we thank the review for pointing out this interesting cancer data set. We will do as the reviewer suggests and intersect our data with Mardakheh Dev Cell 2015 to test the further generality of localisation in neurons and glia, in other cell types. Specifically, we plan to intersect both glial (this study) and neuronal (von Kuegelgen & Chekulaeva, 2020) dataset with protrusive breast cancer cells (Mardakeh et al, 2015).

von Kuegelgen & Chekulaeva, 2020 DOI: 10.1002/wrna.1590

Mardakeh et al, 2015 DOI: 10.1016/j.devcel.2015.10.005

3. The presentation/discussion around Figure 3 is a bit weaker than other parts of the manuscript, and it doesn't really contribute to the story in its current form. Notably there is no discussion about the significance of glia in neurological disorders until the very end of the manuscript (page 21), meaning when its first brought up.. it just sits there as a one off side point. The authors might consider strengthening/tightening up the discussion here, if they really want to keep it as a solo main figure rather than integrating it somewhere else/putting it into supplemental. In my view, Figures 2 & 3 should be merged into something a bit more streamlined.

This is a good point. We plan to strengthen the presentation of Figure 3 and discussion of the significance of glia in neurological disorders by adding a description of the Figure in the Results section and highlighting the significance of glia in nervous system disorders in the Discussion section.

4. Why aren't there more examples of different mRNAs in Figure 4? Seems a waste to kick them all to supplemental.

We agree that it could be helpful to show different expression patterns in the main figure. To address this point we will add Pdi (Fig. S4D), which shows mRNA expression in both the glia and the surrounding muscle cell. This pattern is in contrast to Gs2, which is highly specific to glial cells. We will also note that although *pdi* mRNA is present in both the glia and muscle, Pdi protein is only abundant in the glia, suggesting that translation of *pdi* mRNA to protein is regulated in a cell-specific manner.

5. The plasticity experiments, while creative, I think need to be approached far more cautiously in their interpretation. Given that the siRNAs will completely deplete these mRNAs- it really needs to be stressed any/all of the effects seen could just be the result of "defective" or "altered" states in this glial population- which has spill over effects on plasticity in at the NMJ. Without directly visualizing if these mRNAs are locally translated in these processes and assessing if their translation is modulated by their plasticity paradigm, all these experiments can say is that these RNAs are needed in glia to modulate ghost bouton formation in axons. This represents the weakest part of this manuscript, and the part that I feel does not actually backup the claims currently being made. Without any experiments to A. quantify how much of these transcripts are localized vs in the cell body of these glia, B. visualize/quantify the translation of these mRNAs during baseline and during plasticity; the authors cannot use these data to claim that localized mRNAs are required for synaptic plasticity.

We are grateful to the reviewer for pointing out that we were not precise enough in defining our interpretation of the structural plasticity assay. We did not intend to claim that our results show that local translation of these transcripts is necessary for plasticity, only that these transcripts are localized and are required in the glia for plasticity in the adjacent neuron (in which the transcript levels are not disrupted in the experiment). Definitely proving that these transcripts are required locally and

Revision Plan

translated in response to synaptic activity would require genetic/chemical perturbations and imaging assays that would require a year or more to complete, so are beyond the scope of this manuscript. To address this point, we will clarify that the results do not show that *localized* transcripts are required, only that the transcripts are required somewhere specifically in the glial cell (without affecting the neuron level), and we can indeed show in an independent experiment that there are localized transcripts.

Reviewer #2 -----

Major points:

1. The authors analyse the 1700 shortlisted genes for Gene Ontology and associations with autism spectrum disorder, leading to interesting results. However, it is not clear to what extent the enrichments they observe are driven by their presumptive localization or if the associations are driven to a significant extent by the presence of these genes in the selected cell types in the Fly Cell Atlas. One way to address this would be to perform the GO and SFARI analysis on genes that are expressed in the same cells in the Fly Cell Atlas but were not shortlisted from the mammalian cell datasets - the results could then be compared to those obtained with the 1700 localized transcripts.

This is a fair point raised by the reviewer as genes involved in neurological disease such as Autism Spectrum Disorder may be enriched in CNS/PNS cell types. We will follow the reviewer's suggestion to perform GO and SFARI gene enrichment analysis in genes that were not shortlisted for presumptive glial localisation.

2. Although the authors attempt to justify its inclusion, I'm not convinced why it was important to use the whole cell transcriptome of perisynaptic Schwann cells as part of the selection process for localizing transcripts. Including this dataset may reduce the power of the pipeline by including mRNAs that are not localized to protrusions. How many of the shortlisted 1700 genes, and how many of the 11 glial localized mRNAs in Table 5, would be lost if the whole cell transcriptome were excluded. More generally, what is the distribution of the 11 validated localizing transcripts in each dataset in Table 4? This information might be valuable for determining which dataset(s), if any, has the best predictive power in this context.

We thank the reviewer for raising this point, which we will address with further analysis and adding to the discussion. We propose to address the criticism by running our analysis pipeline without the inclusion of the dataset using Perisynaptic Schwann Cells (PSCs) and then intersect with the PSCs-expressed genes, since their functional similarity with polarised *Drosophila* glial cells is highly relevant. We also agree with the reviewer that it would be a useful control for us to assess the 'predictive power' of each glial dataset by calculating their contribution to the shortlisted 1,700 glial localised transcripts and to the 11 experimentally validated transcripts via *in situ* hybridisation. To address this point, we plan to add this information in the revised manuscript.

3. Did the authors check if any of the RNAi constructs are reducing levels of the target mRNA or

Revision Plan

protein? Doing so would strengthen the confidence in these important results significantly. In any case, the authors should also mention the caveat of potential off-target effects of RNAi.

We thank the reviewer for their useful comment and agree that the extent to which the RNAi expression reduces the levels of mRNA is not specifically known. We will add a FISH experiment on *lac*, *pdi* and *gs2* RNAi showing very strong reduction in mRNA levels. We will also add an explanation of the caveats of the use of the RNAi system to the discussion.

4. Methods: what is the justification for assuming that if the RNAi cross caused embryonic or larval lethality then the 'next most suitable' RNAi line is reporting on a phenotype specific to the gene. If the authors want to claim the effect is associated with different degrees of knockdown they should show this experimentally. An alternative explanation is that the line used for phenotypic analysis in glia is associated with an off-target effect.

We thank the reviewer for this comment. We agree that off target effects cannot in principle be completely ruled out without considerable additional experimental analysis beyond the scope of this manuscript. To address the criticism we will remove the expression data of the lines that cause lethality and revise the discussion to explain that the level of knockdown in each line is unknown, and would require further experimental exploration.

Minor points:

1. It would be helpful to have in the Introduction (rather than the Results, as is currently the case) an operational definition of mRNA localization in the context of the study. And is it known whether or not localization in protrusions is the norm in mammalian glia or the Drosophila larval glia? I ask because it may be that almost all mRNAs diffuse into the protrusion, so this is not a selective process. One interesting approach to test this idea might be to test if the 1700 shortlisted transcripts have a significant underrepresentation of 'housekeeping' functions.

We thank the reviewer for this excellent suggestion. To address the comment, we will move our explanation of the operational definition of mRNA localization to the Introduction. We will also perform enrichment analysis of housekeeping genes within 1,700 shortlisted transcripts compared to the transcriptome background, as the reviewer suggested.

Reviewer #3 -----

Major points:

1. The authors have pooled data from different studies across different type of glial cells performed from in vitro to in vivo. While pooling datasets may reveal common transcripts enriched in processes, this may not be the best approach considering these are completely different types of glial cells with distinct function in neuronal physiology.

Revision Plan

We thank the reviewer for highlighting the need for us to further justify why we pooled datasets. We will revise the manuscript to better emphasise that the overarching goal of our study was to try to discern a common set of localised transcripts shared between the cells. The problem with analysing and comparing individual data sets is that much of the variation may be due to differences in the methods used and amount of material, rather than differences in the type of cells used. We will revise the discussion to make this point and plan to explain that our approach corresponds well with a previous publication pooling localised mRNA datasets in neurons (von Kugelgen & Chekulaeva 2021).

von Kuegelgen & Chekulaeva, 2020 DOI: 10.1002/wrna.1590

2. It is important to note the limitations of the study. For example, DeSeq2 is biased for highly expressed transcripts. How robust was the prediction for low abundance transcripts?

The presented 1,700 transcripts were shortlisted based on their presence and expression level (TPM) in glial protrusions rather than their relative enrichment. Nevertheless, the reviewer makes a valid criticism of our use of DESeq2, where we compared enriched transcripts in glial and neuronal protrusions in Figure 1D. To address this point we will discuss this caveat in the relevant section.

The issue raised regarding low abundance transcript prediction raises an important question: does the likelihood of localisation to cell extremities correlate with mRNA abundance? We have already partially addressed this point, since our analysis of the fraction of localised transcripts per expression level quantiles shows only limited correlation. To address this comment, we will add these results in the revised manuscript as a supplementary figure.

3. The authors identify 1,700 transcripts that they classify as "predicted to be present" in the projections of the Drosophila PNS glia. This was based on the comparison to all the mammalian glial transcripts. Since the authors have access to a transcriptomic study from Perisynaptic Schwann cells (PSCs), the nonmyelinating glia associated with the NMJ isolated from mice; it would be more convincing to then validate the extent of overlap between Drosophila peripheral glial with the mammalian PSCs. This may reveal conserved features of localized transcripts in the PNS, particularly associated with the NMJ function.

Thank you for the valuable suggestion. A similar point was also raised by [Reviewer #2 - Major point 2] to re-run our pipeline excluding the PSCs dataset and intersect with the PSC transcriptome post-hoc. Please see the above section for our detailed response.

4. Fig 2: What is the extent of overlap between the translating fractions versus the localized fraction? It will be informative to perform the functional annotation of the translating glial transcripts as identified from Fig 1D.

This is an interesting question. To address this point, we plan to: (i) compare transcripts that are translated vs. localised in glial protrusions, and (ii) perform functional annotation enrichment analysis on the translated fraction of genes.

5. "We conclude predicted group of 1,700 are highly likely to be peripherally localized in Drosophila cytoplasmic glial projections". To validate their predictions, the authors test some of these candidates in only one glial cell type. It might be worthy to extend this for other differentially expressed genes localized in another glial type as well.

The presented *in vivo* analyses made use of the *repo-GAL4* driver, which is active in all glial subtypes, including subperineurial, perineurial and wrapping glia that make distal projection to the larval neuromuscular junction. We agree that subtype-specific analysis would be highly informative, but we believe this is outside the scope of the current work where we aimed to identify conserved localised transcriptomes across all glial subtypes. Nevertheless, to address the comment, we plan to further clarify our use of pan-glial *repo-GAL4* driver in the Results and Method section of the revised manuscript.

6. Figure 5: The authors perform KD of candidate transcripts to test the effect on synapse formation. However, these are KD with RNAi that spans across the entire cell. To make the claim about the importance of "target" RNA localization in glia stronger, ideally, they should disrupt the enrichment specifically in the glial protusions and test the impact on bouton formation. Do these three RNAs have any putative localization elements?

We agree with the reviewer, that we would ideally test the effect of disruption of mRNA localization (and therefore localised translation). However, we feel these experiments are beyond the scope of this current study, as they will require a long road of defining localisation signals that are small enough to disrupt without affecting other functions. To address this comment we will revise the Discussion section to mention those difficulties explicitly, and clarify the limitations of the approach used in our study for greater transparency.

Reviewer #4 -----

Major points:

1. The authors use FISH to validate the glial expression of their target genes, though these experiments are not quantified, and no controls are shown. The authors should provide a supplemental figure with "no probe" controls, and/or validate the specificity of the probe via glial knockdown of the target gene (see point 2). Furthermore, these data should be quantified (e.g. number of puncta colocalized with NMJ glia membrans).

Thank you for requesting further information regarding the YFP smFISH probes. We have validated the specificity and sensitivity of the YFP probe in our recent publication (Titlow et al, 2023, Figure 1 and S1). Specifically, we demonstrated the lack of YFP probe signal from wild-type untagged biosamples and showed colocalization of YFP spots with additional probes targeting the endogenous exon of the transcript. Nevertheless, we will address this comment by adding control image panels of smFISH in wild-type (*OrR*) neuromuscular junction preparations.

Titlow et al, 2023 DOI: 10.1083/jcb.202205129

2. For the most part, the authors only use one RNAi line for their functional studies, and they only show data for one line, even if multiple were used. To rule out potential false negatives, the authors should leverage their FISH probes to show the efficacy of their knockdowns in glia. This would serve the dual purpose of validating the new probes (see point 1).

Revision Plan

Thank you for the suggestion. This point was also raised by [Reviewer #2 - Major point 3]. Please see above for our detailed response.

3. In Figure 5 E, given the severe reduction in size in the stimulated *Pdi* KD animals, the authors should show images of the unstimulated nerve as well. Do the nerve terminals actually shrink in size in these animals following stimulation, rather than expand? The NMJ looks substantially smaller than a normal L3 NMJ, though their quantification of neurite size in F suggests they're normal until stimulation.

We share the same interpretation of the data with the reviewer that the neurite area is reduced post-potassium stimulation in *pdi* knockdown animals. We will follow the reviewer's suggestion and add an image showing unstimulated neuromuscular junctions.

Minor points:

2. The authors claim that there is an enrichment of ASD-related genes in their final list of ~1400 genes that are enriched in glial processes. It is well-appreciated that synaptically-localized mRNAs are generally linked to ASDs. Can the authors comment on whether the transcripts localized to glial processes are even more linked to ASDs and neurological disorders than transcripts known to be localized to neuronal processes?

This is an interesting point. To address the comment, we will add a comparison of the degree of enrichment of ASD-related genes in neurite vs. glial protrusions in the revised manuscript.

3. Description of the revisions that have already been incorporated in the transferred manuscript

Reviewer #1

1. The use of blue/green or blue/green/magenta is difficult to resolve in some places. Swapping blue for cyan would greatly aid in visualizing their data.

This comment is much appreciated. We have swapped blue for cyan in Figures 4 and S4. We have also changed Figure S1 to increase contrast and visibility as per reviewer's comment.

2. Make the colouring/formatting of the tables more consistent, its distracting when its constantly changing (also there is no need for a blue background.. just use a basic white table).

This comment is much appreciated. We have applied a consistent colour palette to the Tables without background colourings and made the formatting uniform.

Reviewer #2

2. Introduction: 'Asymmetric mRNA localization is likely to be as important in glia, as it is in neurons,...'. Remove commas

Thank you for pointing this mistake out. We have made the corresponding edits.

Reviewer #3

7. RNA localization in oligodendrocytes has been well studied and characterized. The authors should cite and discuss those papers (PMID: 18442491; PMID: 9281585).

We thank the reviewer for this useful suggestion. We have added these references to the paper.

Reviewer #4

1. In Figure 5D, the authors should include a label to indicate that these images are from an unstimulated condition.

We thank the reviewer for pointing this out. We have added the label as requested.

3. The authors are missing a number of key citations for studies that have explored the functional significance of mRNA trafficking in glia, and those that have validated activity-dependent translation:

Revision Plan

- <https://pubmed.ncbi.nlm.nih.gov/18490510/>

- <https://pubmed.ncbi.nlm.nih.gov/7691830/>

- <https://journals.plos.org/plosbiology/article?id=10.1371/journal.pbio.3001053>

- <https://www.ncbi.nlm.nih.gov/pmc/articles/PMC7450274/>

- <https://pubmed.ncbi.nlm.nih.gov/36261025/>

We thank the reviewer for the comment. We have added these references to the text.

4. Description of analyses that authors prefer not to carry out

July 16, 2023

Re: JCB manuscript #202306152T

Prof. Ilan Davis
University of Oxford
Department of Biochemistry
South Parks Road
Oxford OX1 3QU
United Kingdom

Dear Prof. Davis,

Thank you for submitting your manuscript entitled "Mammalian glial protrusion transcriptomes predict localization of Drosophila glial transcripts required for synaptic plasticity." After assessing the reports from Review Commons and your revision plan we invite you to submit a revision for the JCB Tool format.

We agree that quantifying peripheral enrichment and translation as well as identifying peripheral localization targeting sequences is beyond the scope of this study. However, we do feel that it is important to validate the RNAi mediated depletion for all ten assayed genes rather than just the three mentioned in the revision plan. If direct confirmation of depletion is not possible for some reason then replication of the results with non-overlapping RNAis would also be acceptable. Additionally, although identifying new transcripts that are peripherally localized is not required for resubmission, any such data would significantly enhance the work and we encourage you to add it to the paper if possible.

GENERAL GUIDELINES:

Text limits: Character count for an Transfer is < 40,000, not including spaces. Count includes title page, abstract, introduction, results, discussion, and acknowledgments. Count does not include materials and methods, figure legends, references, tables, or supplemental legends.

Figures: Transfers may have up to 10 main text figures. Figures must be prepared according to the policies outlined in our Instructions to Authors, under Data Presentation, <https://jcb.rupress.org/site/misc/ifora.xhtml>. All figures in accepted manuscripts will be screened prior to publication.

Title: The title should be less than 100 characters including spaces. Make the title concise but accessible to a general readership.

*****IMPORTANT:** It is JCB policy that if requested, original data images must be made available. Failure to provide original images upon request will result in unavoidable delays in publication. Please ensure that you have access to all original microscopy and blot data images before submitting your revision. ***

Supplemental information: There are strict limits on the allowable amount of supplemental data. Tool papers may have up to 5 supplemental figures. Up to 10 supplemental videos or flash animations are allowed. A summary of all supplemental material should appear at the end of the Materials and methods section.

Please note that JCB now requires authors to submit Source Data used to generate figures containing gels and Western blots with all revised manuscripts. This Source Data consists of fully uncropped and unprocessed images for each gel/blot displayed in the main and supplemental figures. If your paper will include cropped gel and/or blot images, please be sure to provide one Source Data file for each figure that contains gels and/or blots along with your revised manuscript files. File names for Source Data figures should be alphanumeric without any spaces or special characters (i.e., SourceDataF#, where F# refers to the associated main figure number or SourceDataFS# for those associated with Supplementary figures). The lanes of the gels/blots should be labeled as they are in the associated figure, the place where cropping was applied should be marked (with a box), and molecular weight/size standards should be labeled wherever possible. Source Data files will be made available to reviewers during evaluation of revised manuscripts and, if your paper is eventually published in JCB, the files will be directly linked to specific figures in the published article.

The typical timeframe for revisions is three to four months. While most universities and institutes have reopened labs and allowed researchers to begin working at nearly pre-pandemic levels, we at JCB realize that the lingering effects of the COVID-19 pandemic may still be impacting some aspects of your work, including the acquisition of equipment and reagents. Therefore, if you anticipate any difficulties in meeting this aforementioned revision time limit, please contact us and we can work with you to find an appropriate time frame for resubmission. Please note that papers are generally considered through only one revision cycle, so any revised manuscript will likely be either accepted or rejected.

Thank you for this interesting contribution to Journal of Cell Biology. You can contact us at the journal office with any questions, cellbio@rockefeller.edu or call (212) 327-8588.

Sincerely,

Marc Freeman, PhD
Monitoring Editor
Journal of Cell Biology

Dan Simon, PhD
Scientific Editor
Journal of Cell Biology

Full Revision

Manuscript number: 202306152TR
Corresponding author(s): Ilan, Davis

1. General Statements

We thank the reviewers for their well informed and constructive comments. We have addressed every comment in detail, as explained below, and have revised the manuscript accordingly. We feel that the reviewing process has been excellent and has improved the manuscript considerably either by the addition of considerable new data or by revisions to the text. In addition to a clean revised manuscript, a marked up version of the manuscript has been submitted to indicate where we have made changes to the text and figures (highlighted yellow). We apologize for the delay in resubmission of the revised manuscript as we have been heavily affected by the lab's relocation from Oxford to Glasgow University. The authors' affiliations have been updated in the revised manuscript accordingly.

In summary, here are some of the key improvements we feel we have made in response to the reviewers comments and suggestions:

- A) Modification of our meta-analysis pipeline to remove the Perisynaptic Schwann cell dataset.
- B) Addition of breast cancer cell protrusion RNA-seq dataset.
- C) Validation of all 10 *UAS-RNAi* lines used in this study using smFISH.
- D) Validation of the *YFP* smFISH probe set used in this study.
- E) New GO/disease-ontology analysis comparing predicted localized vs. non-localized genes.
- F) New data showing underrepresentation of housekeeping genes in glial localized transcripts.
- G) Clarification of our definition of RNA localization and interpretation of plasticity experiments.
- H) Modification of figure panels to improve visibility.
- I) Addition of relevant references suggested by the reviewers.

Reviewer #1 -----

Major points:

1. The presented work itself (Figures 1-4) does not need significant adjustments prior to publication, in my view, with only a few points to address. However, the work in Figure 5- doesn't really support the claims the authors make on its own, and would require some additional experiments or at the very least discussion of the caveats to its current form.

This is a very good point, made by this reviewer here and also below (comment 5) and by reviewer #3 (comment 6). Namely, that our use of RNAi knockdown in the glia demonstrates that the transcripts are required for structural plasticity, not that their localisation is specifically required. The reviewer is implying that if one knew the localisation signal, then one could specifically disrupt the localisation to

test specifically whether the localisation itself in glia is required for structural plasticity in the motoneuron. This is precisely what we are planning to do in the longer term, but such experiments are difficult and novel, so are well beyond the scope of this manuscript. The experiments would require first mapping the localisation signals and then making very novel constructs to knock out the localisation signal in a conditional manner, so that protein was only made from mRNA in the cell body, not in the periphery. To address this excellent point, we have explained the caveat in the discussion, and also added to the discussion suggestions of how future experiments could test the functional importance of localized mRNA translation in the glia to neuronal plasticity. The following statement has been added in the discussion [page 12]:

Our results do not explicitly test whether the localization of the mRNA in the glial periphery is specifically required for the genes to function in the glia and to influence plasticity in the adjacent motoneurons. Showing that would require future experiments to map the localization signals and then knocking out localization in a conditional manner, so that the mRNAs are only translated in the cell body and not in the cytoplasmic extensions of the glia. Future follow up experiments could also include Sun-Tag (Tanenbaum et al., 2014) and FLARIM (Richer et al., 2021) to visualize local translation directly.

2. Localized glia transcripts, are they "glial/CNS/PNS" significant or are they similar to other known datasets of protrusion transcriptomes? The authors compared their 4801 "total" localized to a local transcriptome dataset from the Chekulaeva lab finding that a significant fraction are localized in both. As the authors note, this is in good agreement with a recent paper from the Talifarro lab showing conservation of localization of mRNAs across different cell types. What the authors haven't done here, is further test this by looking at other non-neuronal projection transcriptomic datasets (for example Mardakheh Developmental Cell 2015, among others). If the predicted glia-localized processes are similar to non-neuronal processes transcriptomes, this would further strengthen this claim and rule out some level of CNS/PNS derived lineage driving the similarities between glia and neuronal localized transcripts.

Thank you for pointing out this interesting cancer data set. We have intersected our glial dataset with the neurite (von Kuegelgen & Chekulaeva, 2020) and breast cancer cell protrusion localized RNAs (Mardakheh et al, 2015). To address this point, we have done the comparisons the reviewer suggested. Interestingly, we found significant overlap of localized transcripts between neural (glial/neurite) and non-neural cells (Figure 1E & [page 5]). These additional results add further evidence of a conserved mechanism of RNA localisation that transcends cell morphology and type (Goering et al, 2022).

3. The presentation/discussion around Figure 3 is a bit weaker than other parts of the manuscript, and it doesn't really contribute to the story in its current form. Notably there is no discussion about the significance of glia in neurological disorders until the very end of the manuscript (page 21), meaning when its first brought up.. it just sits there as a one off side point. The authors might consider strengthening/tightening up the discussion here, if they really want to keep it as a solo main figure rather than integrating it somewhere else/putting it into supplemental. In my view, Figures 2 & 3 should be merged into something a bit more streamlined.

Thank you for the suggestion. To address this criticism, we have combined GO and reactome enrichment analyses into a single figure to convey a streamlined message (Figure 3). We further strengthened our findings by performing a comparative GO enrichment between localized versus non-localized

transcripts. This analysis revealed enrichment of distinct biological functions between localized and non-localized transcripts. The additional comparative analysis is presented in the **Figure S2A, S2C & [page 6]**.

We also reinforced the connection between glial-localized transcripts and neurological disease by showing under-representation of disease ontology association of non-localized transcripts. We also show non-localized transcripts overlap with the SFARI list to a lesser extent than localized RNAs. These analyses are presented in the new **Figure 4A-C**, and we have expanded the corresponding Results section to better explain our results and their significance **[page 7]**.

4. Why aren't there more examples of different mRNAs in Figure 4? Seems a waste to kick them all to supplemental.

We agree that it could be helpful to show additional examples of RNA localisation patterns in the main figure. To address this point, we have added Pdi (from previous Figure S4D) to the main **Figure 5F**, which shows mRNA expression in both the glia and the surrounding muscle cell. This pattern is in contrast to Gs2 (**Figure 5A**), which is highly specific to glial cells. We have also added the following statement in the Results section **[page 9]**:

Interestingly, although Pdi mRNA is present in both glia and muscle, Pdi::YFP protein is only abundant in the glia (Figure 5F-I), indicating that translation of Pdi mRNA to protein is regulated in a cell specific manner.

5. The plasticity experiments, while creative, I think need to be approached far more cautiously in their interpretation. Given that the siRNAs will completely deplete these mRNAs- it really needs to be stressed any/all of the effects seen could just be the result of "defective" or "altered" states in this glial population- which has spill over effects on plasticity in at the NMJ. Without directly visualizing if these mRNAs are locally translated in these processes and assessing if their translation is modulated by their plasticity paradigm, all these experiments can say is that these RNAs are needed in glia to modulate ghost bouton formation in axons. This represents the weakest part of this manuscript, and the part that I feel does not actually backup the claims currently being made. Without any experiments to A. quantify how much of these transcripts are localized vs in the cell body of these glia, B. visualize/quantify the translation of these mRNAs during baseline and during plasticity; the authors cannot use these data to claim that localized mRNAs are required for synaptic plasticity.

We are grateful to the reviewer for further pointing out that we were not precise enough in defining our interpretation of the structural plasticity assay. We did not intend to claim that our results show that local translation of these transcripts is necessary for plasticity, only that these transcripts are localized and are required in the glia for plasticity in the adjacent neuron, in which the transcript levels remain unchanged in the experiment. Definitively proving that these transcripts are required locally and translated locally in response to synaptic activity would require a lengthy project that is certainly beyond the scope of this manuscript. Nevertheless, to address the comment, we explain this point in the discussion (see comment 1). We also briefly mention in the discussion that future experiments could include methods such as the quantification of the mRNA foci and the disruption of the mRNA localisation signals to disrupt localized translation and the use of techniques such as *Sun-Tag* (Tanenbaum et al, 2014) and *FLARIM* (Richer et al, 2021) to visualize local translation directly **(added to [page 12])**.

Full Revision

Minor points:

1. The use of blue/green or blue/green/magenta is difficult to resolve in some places. Swapping blue for cyan would greatly aid in visualizing their data.

We swapped blue for cyan in **Figures 5 and S3**. We also changed **Figure S1** to increase contrast and visibility as per reviewer's comment.

2. Make the colouring/formatting of the tables more consistent, its distracting when its constantly changing (also there is no need for a blue background.. just use a basic white table).

We have applied a consistent color palette to the Tables without background colourings and made the formatting uniform.

Reviewer #2 -----

Major points:

1. The authors analyse the 1700 shortlisted genes for Gene Ontology and associations with autism spectrum disorder, leading to interesting results. However, it is not clear to what extent the enrichments they observe are driven by their presumptive localization or if the associations are driven to a significant extent by the presence of these genes in the selected cell types in the Fly Cell Atlas. One way to address this would be to perform the GO and SFARI analysis on genes that are expressed in the same cells in the Fly Cell Atlas but were not shortlisted from the mammalian cell datasets - the results could then be compared to those obtained with the 1700 localized transcripts.

This is a fair point raised by the reviewer. To address this, we have followed the reviewer's suggestion to perform comparative GO enrichment analysis between localized and non-localized transcripts. We found functionally distinct groups of GO terms were enriched in localized vs. non-localized genes (**Figure S2A**). We also performed Reactome analysis of non-localized genes and we found a lack of their association with RNA metabolism, immune response, and membrane transport terms (**Figure S2C & [page 7]**). Furthermore, we found that non-localized transcripts overlap with the SFARI genes at a much lesser degree than the localized transcripts (**Figure 4C & [page 6]**). We believe these comparisons reinforce the unique functional characteristics of the glial protrusion localized transcripts as opposed to the general cell type function.

2. Although the authors attempt to justify its inclusion, I'm not convinced why it was important to use the whole cell transcriptome of perisynaptic Schwann cells as part of the selection process for localizing transcripts. Including this dataset may reduce the power of the pipeline by including mRNAs that are not localized to protrusions. How many of the shortlisted 1700 genes, and how many of the 11 glial localized mRNAs in Table 5, would be lost if the whole cell transcriptome were excluded. More generally, what is the distribution of the 11 validated localizing transcripts in each dataset in Table 4? This information might be valuable for determining which dataset(s), if any, has the best predictive power in this context.

We thank the reviewer for raising this point. We have addressed the criticism by re-running our analysis pipeline without the whole Perisynaptic Schwann Cells (PSCs) libraries, resulting in a new 1,740 set (**Figure 1F**). Importantly, we recovered all 11 validated localizing transcripts after the re-analysis. We then intersected the 1,740 genes with the PSCs-expressed genes, since their functional similarity with polarized *Drosophila* glial cells is highly relevant (**Figure 2A & [page 6]**). Our post-hoc analysis supported that 97% of the 1,740 genes are expressed in the PSC cell type, which strengthens the cell type relevance of our localisation prediction.

We also calculated the contribution of each glial dataset to the 11 validated transcripts (**Figure S3J**). Most of the mammalian libraries supported evidence of localisation of the 11 genes to glial periphery, which supports ubiquitousness of localized transcripts amongst glial subtypes. However, we did not find differential predictive power between the mammalian libraries that was statistically significant. We have added following statement accompanying the new figure [**page 9**]:

We found the RNA localization of these 11 transcripts were supported by most mammalian libraries (Figure S3J), suggesting evolutionary conservation of RNA localization.

3. Did the authors check if any of the RNAi constructs are reducing levels of the target mRNA or protein? Doing so would strengthen the confidence in these important results significantly. In any case, the authors should also mention the caveat of potential off-target effects of RNAi.

We thank the reviewer for their useful comment and agree that the extent to which the RNAi expression reduces the levels of mRNA is not specifically known. We have added smFISH images for all 10 genes used in the RNAi experiment where we examined the knockdown using smFISH probes targeting endogenous transcripts (**Figure S5**). We show strong reductions in mRNA levels in RNAi conditions.

We also agree that off-target effects cannot in principle be completely ruled out without considerable additional experimental analysis beyond the scope of this manuscript. To address this point, we have mentioned this caveat in the Discussion as below [**page 13**]:

Whilst we used GAL4-UAS-RNAi system to knockdown target transcripts and validated the knockdowns using smFISH, we cannot completely exclude potential off-target effects of the RNAi system and future studies should consider orthogonal mutant genotypes for detailed mechanical analysis.

4. Methods: what is the justification for assuming that if the RNAi cross caused embryonic or larval lethality then the 'next most suitable' RNAi line is reporting on a phenotype specific to the gene. If the authors want to claim the effect is associated with different degrees of knockdown they should show this experimentally. An alternative explanation is that the line used for phenotypic analysis in glia is associated with an off-target effect.

We thank the reviewer for this comment. To fully address this comment, we first revised the Methods section to clarify our experimental procedures, and most importantly, we have now added data to validate all 10 RNAi lines used in this study using smFISH showing a strong reductions in expression levels (**Figure S5**). The following statement has been added to the Methods section [**page 16**]:

We selected viable 3rd instar larvae for RNAi-based phenotypic assays, and we validated gene knockdowns by visualizing transcript expression in the NMJ glia. We did not recover any viable 3rd instar

Full Revision

larvae after glial-specific knockdown of Cip4, which was excluded from further analysis of synaptic plasticity phenotypes.

For the point regarding caveats of the RNAi system, please see our detailed response above [Reviewer #2 - Major point 3].

Minor points:

1. It would be helpful to have in the Introduction (rather than the Results, as is currently the case) an operational definition of mRNA localization in the context of the study. And is it known whether or not localization in protrusions is the norm in mammalian glia or the Drosophila larval glia? I ask because it may be that almost all mRNAs diffuse into the protrusion, so this is not a selective process. One interesting approach to test this idea might be to test if the 1700 shortlisted transcripts have a significant underrepresentation of 'housekeeping' functions.

We thank the reviewer for this excellent suggestion. To address the comment, we have moved our explanation of the operational definition of mRNA localization to the Introduction. We have also performed enrichment analysis of housekeeping genes (Joshi et al, 2022 PloS One) within 1,740 shortlisted transcripts compared to the background, as the reviewer suggested. To this end, we found significant under-representation of housekeeping functions in the 1,740 genes (Figure 2B & [page 6]), which suggests functional selectivity of localized RNAs.

2. Introduction: 'Asymmetric mRNA localization is likely to be as important in glia, as it is in neurons,...'. Remove commas

Thank you for pointing this mistake out. We have made the corresponding edits.

Reviewer #3 -----

Major points:

1. The authors have pooled data from different studies across different type of glial cells performed from in vitro to in vivo. While pooling datasets may reveal common transcripts enriched in processes, this may not be the best approach considering these are completely different types of glial cells with distinct function in neuronal physiology.

We thank the reviewer for highlighting the need for us to further justify why we pooled datasets. In response, we have revised the manuscript to better emphasize that the overarching goal of our study was to try to discern a common set of localized transcripts shared between glial cells. The challenge with analyzing and comparing individual datasets is that much of the variation may be due to differences in the methods used and amount of starting material, rather than differences in the cell type. We have revised the Introduction and the Discussion to make this point clear and explained that our approach

Full Revision

corresponds well with a previous publication pooling localized mRNA datasets in neurites (von Kugelgen & Chekulaeva 2021). The relevant section in the Introduction has been revised as follows [page 2]:

These varying glial morphologies and functional roles are suggestive of mRNA localization to glial protrusions, and indeed some research has outlined its importance (Blanco-Urrejola et al., 2021; Meservey et al., 2021). However, there has yet to be a systematic study comparing glial protrusion datasets to assess whether mRNA localization is pervasive in glia or the extent to which localized transcripts overlap in the periphery of different glial subtypes or in neurons.

The following statement has been added in the Discussion section [page 12]:

Here, we performed meta-analysis of multiple published datasets of peripherally localized mammalian glial mRNAs to identify the core glial protrusion transcriptome shared across multiple glial subtypes. Our comprehensive analysis corresponds well with the previously published meta-analysis of neurite transcriptomes (von Kugelgen & Chekulaeva, 2020) and it provides a valuable resource for future studies on glial mRNA localization.

2. It is important to note the limitations of the study. For example, DeSeq2 is biased for highly expressed transcripts. How robust was the prediction for low abundance transcripts?

The presented 1,740 transcripts were shortlisted based on their presence and expression level (TPM) in glial protrusions rather than their relative enrichment. Nevertheless, the reviewer makes a valid criticism of the limitation of DESeq2 in the previous Figure 1D, and we agree that the enrichment analysis can be biased and highly variable between datasets. To address this point, we moved our operational definition of RNA localisation to the Introduction and removed DESeq2-associated outputs from this study to streamline our analysis pipeline.

The issue raised regarding the low abundance transcript prediction raises an important question: does the likelihood of localisation to cell extremities correlate with mRNA abundance? We have addressed this point by analyzing the fraction of localized transcripts per expression level quantiles, which showed limited correlation. This new analysis has been added in the new **Figure 2B-C**.

3. The authors identify 1,700 transcripts that they classify as "predicted to be present" in the projections of the Drosophila PNS glia. This was based on the comparison to all the mammalian glial transcripts. Since the authors have access to a transcriptomic study from Perisynaptic Schwann cells (PSCs), the nonmyelinating glia associated with the NMJ isolated from mice; it would be more convincing to then validate the extent of overlap between Drosophila peripheral glial with the mammalian PSCs. This may reveal conserved features of localized transcripts in the PNS, particularly associated with the NMJ function.

Thank you for the valuable suggestion. A similar point was also raised by [Reviewer #2 - Major point 2], and we have addressed this by re-running our pipeline excluding the PSCs dataset and intersecting the resulting set with the PSC transcriptome post-hoc. Please see the above section for our detailed response.

4. Fig 2: What is the extent of overlap between the translating fractions versus the localized fraction? It will be informative to perform the functional annotation of the translating glial transcripts as identified from Fig 1D.

This is an interesting question. To address this point we have identified transcripts that are localized & translated in glial protrusions (~700 genes) and performed comparative GO analysis on the translated fraction of genes versus the localized genes (**Figure S2B**). However, our analysis showed a similar set of GO terms enriched for both localized and translated fractions, which yields limited evidence towards functional discrimination of locally translated transcripts. The following has been added in the Results section accompanying the new figure [page 6]:

Then, we also asked whether locally translated transcripts could be functionally distinct from localized RNAs. To address this question, we further filtered localized transcripts for evidence of translation in glial protrusions in all 4 TRAP-seq libraries in mammals, resulting in 778 genes (3 Astrocytes and 1 microglia, Figure 1B). We found a comparable distribution of enriched GO terms for localized versus localized & translated genes (Figure S2B), suggesting the subset of locally translated transcripts are functionally similar to the localized RNAs.

5. "We conclude predicted group of 1,700 are highly likely to be peripherally localized in *Drosophila* cytoplasmic glial projections". To validate their predictions, the authors test some of these candidates in only one glial cell type. It might be worthy to extend this for other differentially expressed genes localized in another glial type as well.

Our *in vivo* analyses made use of the *Repo-GAL4* driver, which is active in all glial subtypes, including subperineurial, perineurial and wrapping glia that make distal projection to the larval neuromuscular junction. We agree that subtype-specific analysis would be highly informative, but we believe this is outside the scope of the current work where we aimed to identify conserved localized transcriptomes across all glial subtypes. Nevertheless, to address the comment, we have further clarified our use of pan-glial *repo-GAL4* driver in the Methods section and the following statement has been added in the Results section [page 9]:

We used UAS-RNAi lines against 11 different transcripts driven by the pan-glial Repo-GAL4 driver, which is active in all three glial subtypes that make distal cell projection to NMJ motoneuron neurites (Figure 1 and Figure S1A-C).

6. Figure 5: The authors perform KD of candidate transcripts to test the effect on synapse formation. However, these are KD with RNAi that spans across the entire cell. To make the claim about the importance of "target" RNA localization in glia stronger, ideally, they should disrupt the enrichment specifically in the glial protrusions and test the impact on bouton formation. Do these three RNAs have any putative localization elements?

This is a good point, also made in two other comments above. We fully addressed this comment in the discussion section, as explained above in the responses to comments [Reviewer #1 - Major points 1 & 5].

7. RNA localization in oligodendrocytes has been well studied and characterized. The authors should cite and discuss those papers (PMID: 18442491; PMID: 9281585).

We thank the reviewer for this useful suggestion and we added these references to the paper.

Full Revision

Reviewer #4 -----

Major points:

1. The authors use FISH to validate the glial expression of their target genes, though these experiments are not quantified, and no controls are shown. The authors should provide a supplemental figure with "no probe" controls, and/or validate the specificity of the probe via glial knockdown of the target gene (see point 2). Furthermore, these data should be quantified (e.g. number of puncta colocalized with NMJ glia membrans).

Thank you for requesting further information regarding the YFP smFISH probes. We have addressed this comment by adding control image panels of smFISH in wild-type neuromuscular junction preparations, showing lack of probe-specific signal (**Figure S4**). Following statement has been added in the revised manuscript [**page 9**]:

To this end, we visualized localized transcripts of venusYFP-containing mRNA in the extended NMJ glial protrusions of the YFP insertion lines in each gene (Lowe et al., 2014). The smFISH probe set against venusYFP was validated by showing the lack of fluorescent signal in untagged wildtype-equivalent flies (Figure S4).

We further revised the manuscript to point out in the results section that [**page 9**]:

This result is consistent with our previous validation of the specificity and sensitivity of the YFP probe (Titlow et al., 2023).

2. For the most part, the authors only use one RNAi line for their functional studies, and they only show data for one line, even if multiple were used. To rule out potential false negatives, the authors should leverage their FISH probes to show the efficacy of their knockdowns in glia. This would serve the dual purpose of validating the new probes (see point 1).

Thank you for the suggestion. This point regarding the validation of RNAi lines was also raised by [**Reviewer #2 - Major point 3**]. Please see above for our detailed response.

3. In Figure 5 E, given the severe reduction in size in the stimulated Pdi KD animals, the authors should show images of the unstimulated nerve as well. Do the nerve terminals actually shrink in size in these animals following stimulation, rather than expand? The NMJ looks substantially smaller than a normal L3 NMJ, though their quantification of neurite size in F suggests they're normal until stimulation.

We share the same interpretation of the data with the reviewer that the neurite area is reduced post-potassium stimulation in *pdi* knockdown animals. We have followed the reviewer's suggestion and added image panels showing unstimulated neuromuscular junctions in **Figure 6E**.

Minor points:

1. In Figure 5D, the authors should include a label to indicate that these images are from an unstimulated condition.

Full Revision

We thank the reviewer for pointing this out, and have added the label as requested.

2. The authors claim that there is an enrichment of ASD-related genes in their final list of ~1400 genes that are enriched in glial processes. It is well-appreciated that synaptically-localized mRNAs are generally linked to ASDs. Can the authors comment on whether the transcripts localized to glial processes are even more linked to ASDs and neurological disorders than transcripts known to be localized to neuronal processes?

To address the comment, we have added a comparison of the degree of enrichment of ASD-related genes in neurite vs. glial protrusions in the revised manuscript (**Figure 4D & [page 7]**). We found the significance of the overlap with the SFARI list was lower with neurite RNAs versus glial-protrusion RNAs, which may be attributable to a relatively more diverse neurite transcriptome.

3. The authors are missing a number of key citations for studies that have explored the functional significance of mRNA trafficking in glia, and those that have validated activity-dependent translation:

- <https://pubmed.ncbi.nlm.nih.gov/18490510/>

- <https://pubmed.ncbi.nlm.nih.gov/7691830/>

- <https://journals.plos.org/plosbiology/article?id=10.1371/journal.pbio.3001053>

- <https://www.ncbi.nlm.nih.gov/pmc/articles/PMC7450274/>

- <https://pubmed.ncbi.nlm.nih.gov/36261025/>

We thank the reviewer for their comment, and we have added these references to the text.

June 22, 2024

RE: JCB Manuscript #202306152R

Prof. Ilan Davis
University of Glasgow
School of Molecular Biosciences
University Avenue
University of Glasgow
Glasgow G12 8QQ
United Kingdom

Dear Prof. Davis,

Thank you for submitting your revised manuscript entitled "Murine glial protrusion transcripts predict localized *Drosophila* glial mRNAs involved in plasticity." We would be happy to publish your paper in JCB pending final revisions necessary to meet our formatting guidelines (see details below).

A. MANUSCRIPT ORGANIZATION AND FORMATTING:

1) Text limits: Character count for Tools is < 40,000, not including spaces. Count includes title page, abstract, introduction, results, discussion, and acknowledgments. Count does not include materials and methods, figure legends, references, tables, or supplemental legends.

2) Figure formatting: Tools may have up to 10 main text figures. Scale bars must be present on all microscopy images, including inset magnifications. Please avoid pairing red and green for images and graphs to ensure legibility for color-blind readers. If red and green are paired for images, please ensure that the particular red and green hues used in micrographs are distinctive with any of the colorblind types. If not, please modify colors accordingly or provide separate images of the individual channels.

3) Statistical analysis: Error bars on graphic representations of numerical data must be clearly described in the figure legend. The number of independent data points (n) represented in a graph must be indicated in the legend. Please, indicate whether 'n' refers to technical or biological replicates (i.e. number of analyzed cells, samples or animals, number of independent experiments). If independent experiments with multiple biological replicates have been performed, we recommend using distribution-reproducibility SuperPlots (please see Lord et al., JCB 2020) to better display the distribution of the entire dataset, and report statistics (such as means, error bars, and P values) that address the reproducibility of the findings.

Statistical methods should be explained in full in the materials and methods. For figures presenting pooled data the statistical measure should be defined in the figure legends. Please also be sure to indicate the statistical tests used in each of your experiments (both in the figure legend itself and in a separate methods section) as well as the parameters of the test (for example, if you ran a t-test, please indicate if it was one- or two-sided, etc.). Also, if you used parametric tests, please indicate if the data distribution was tested for normality (and if so, how). If not, you must state something to the effect that "Data distribution was assumed to be normal but this was not formally tested."

4) Materials and methods: Should be comprehensive and not simply reference a previous publication for details on how an experiment was performed. Please provide full descriptions (at least in brief) in the text for readers who may not have access to referenced manuscripts. The text should not refer to methods "...as previously described."

5) For all cell lines, vectors, constructs/cDNAs, etc. - all genetic material: please include database / vendor ID (e.g., Addgene, ATCC, etc.) or if unavailable, please briefly describe their basic genetic features, even if described in other published work or gifted to you by other investigators (and provide references where appropriate). Please be sure to provide the sequences for all of your oligos: primers, si/shRNA, RNAi, gRNAs, etc. in the materials and methods. You must also indicate in the methods the source, species, and catalog numbers/vendor identifiers (where appropriate) for all of your antibodies, including secondary. If antibodies are not commercial, please add a reference citation if possible.

6) Microscope image acquisition: The following information must be provided about the acquisition and processing of images:
a. Make and model of microscope

- b. Type, magnification, and numerical aperture of the objective lenses
- c. Temperature
- d. Imaging medium
- e. Fluorochromes
- f. Camera make and model
- g. Acquisition software
- h. Any software used for image processing subsequent to data acquisition. Please include details and types of operations involved (e.g., type of deconvolution, 3D reconstitutions, surface or volume rendering, gamma adjustments, etc.).

7) References: There is no limit to the number of references cited in a manuscript. References should be cited parenthetically in the text by author and year of publication. Abbreviate the names of journals according to PubMed.

8) Supplemental materials: Tools may have up to 5 supplemental figures and 10 videos. You currently exceed this limit but, in this case, we will be able to give you the extra space. Please also note that tables, like figures, should be provided as individual, editable files. A summary of all supplemental material should appear at the end of the Materials and methods section. Please include one brief sentence per item.

9) eTOC summary: A ~40-50 word summary that describes the context and significance of the findings for a general readership should be included on the title page. The statement should be written in the present tense and refer to the work in the third person. It should begin with "First author name(s) et al..." to match our preferred style.

10) Conflict of interest statement: JCB requires inclusion of a statement in the acknowledgements regarding competing financial interests. If no competing financial interests exist, please include the following statement: "The authors declare no competing financial interests." If competing interests are declared, please follow your statement of these competing interests with the following statement: "The authors declare no further competing financial interests."

11) A separate author contribution section is required following the Acknowledgments in all research manuscripts. All authors should be mentioned and designated by their first and middle initials and full surnames. We encourage use of the CRediT nomenclature (<https://casrai.org/credit/>).

12) ORCID IDs: ORCID IDs are unique identifiers allowing researchers to create a record of their various scholarly contributions in a single place. Please note that ORCID IDs are required for all authors. At resubmission of your final files, please be sure to provide your ORCID ID and those of all co-authors.

13) Journal of Cell Biology now requires a data availability statement for all research article submissions. These statements will be published in the article directly above the Acknowledgments. The statement should address all data underlying the research presented in the manuscript. Please visit the JCB instructions for authors for guidelines and examples of statements at (<https://rupress.org/jcb/pages/editorial-policies#data-availability-statement>).

B. FINAL FILES:

Additionally, JCB encourages authors to submit a short video summary of their work. These videos are intended to convey the main messages of the study to a non-specialist, scientific audience. Think of them as an extended version of your abstract, or a short poster presentation. We encourage first authors to present the results to increase their visibility. The videos will be shared

on social media to promote your work. For more detailed guidelines and tips on preparing your video, please visit <https://rupress.org/jcb/pages/submission-guidelines#videoSummaries>.

Thank you for your attention to these final processing requirements. Please revise and format the manuscript and upload materials within 7 days. If you need an extension for whatever reason, please let us know and we can work with you to determine a suitable revision period.

Thank you for this interesting contribution, we look forward to publishing your paper in Journal of Cell Biology.

Sincerely,

Marc Freeman, PhD
Monitoring Editor
Journal of Cell Biology

Dan Simon, PhD
Scientific Editor
Journal of Cell Biology